# A circular zone of attachment to the extracellular matrix provides directionality to the motility of *Toxoplasma gondii* in 3D

Rachel V Stadler[1], Shane R Nelson[2], David M Warshaw[2], Gary E Ward[1]*

[1]Department of Microbiology and Molecular Genetics, University of Vermont Larner College of Medicine, Burlington, United States; [2]Department of Molecular Physiology and Biophysics, University of Vermont Larner College of Medicine, Burlington, United States

*For correspondence:
Gary.Ward@uvm.edu

Competing interest: The authors declare that no competing interests exist.

**Abstract** *Toxoplasma gondii* is a protozoan parasite that infects 30–40% of the world's population. Infections are typically subclinical but can be severe and, in some cases, life threatening. Central to the virulence of *T. gondii* is an unusual form of substrate-dependent motility that enables the parasite to invade cells of its host and to disseminate throughout the body. A hetero-oligomeric complex of proteins that functions in motility has been characterized, but how these proteins work together to drive forward motion of the parasite remains controversial. A key piece of information needed to understand the underlying mechanism(s) is the directionality of the forces that a moving parasite exerts on the external environment. The linear motor model of motility, which has dominated the field for the past two decades, predicts continuous anterior-to-posterior force generation along the length of the parasite. We show here using three-dimensional traction force mapping that the predominant forces exerted by a moving parasite are instead periodic and directed in toward the parasite at a fixed circular location within the extracellular matrix. These highly localized forces, which are generated by the parasite pulling on the matrix, create a visible constriction in the parasite's plasma membrane. We propose that the ring of inward-directed force corresponds to a circumferential attachment zone between the parasite and the matrix, through which the parasite propels itself to move forward. The combined data suggest a closer connection between the mechanisms underlying parasite motility and host cell invasion than previously recognized. In parasites lacking the major surface adhesin, TgMIC2, neither the inward-directed forces nor the constriction of the parasite membrane are observed. The trajectories of the TgMIC2-deficient parasites are less straight than those of wild-type parasites, suggesting that the annular zone of TgMIC2-mediated attachment to the extracellular matrix normally constrains the directional options available to the parasite as it migrates through its surrounding environment.

## Editor's evaluation

The authors report the biophysics behind *Toxoplasma gondii* locomotion on 3D matrixes that are fluorescently labelled and hence allow the detection of their displacements and calculation of force vectors. The authors discover that tachyzoites move by a high degree of continuous constrictions reminiscent of those seen during cell invasion. They probe not only wild type parasites but also two key mutants, which reveal a striking absence of the constrictions and changed trajectories. Due to the uniqueness of eukaryotic gliding motility, its high speed and the importance of infection, this manuscript will be of general interest not only to cell biologists studying cell migration in context of infectious diseases but also appealing to biophysicists looking at cellular force generation.

## Introduction

Parasites of the phylum Apicomplexa cause pervasive human disease and are responsible for approximately one million deaths annually (*World Health Organization, 2021*; *Sow et al., 2016*). *Toxoplasma gondii* is the most prevalent apicomplexan parasite, infecting 30–40% of the human population worldwide (*Bigna et al., 2020*; *Flegr et al., 2014*; *Pappas et al., 2009*). *T. gondii* infections are typically subclinical in immunocompetent individuals but may cause blindness (*Rothova, 1993*; *Su et al., 2014*). In immunocompromised individuals and the developing fetus, the disease can be life-threatening (*Pappas et al., 2009*; *Elsheikha et al., 2021*; *McLeod et al., 2014*).

*T. gondii* is a single-cell protozoan, and the pathogenesis associated with acute infection is caused by the highly motile tachyzoite life-cycle stage. Like other apicomplexan parasites, the *T. gondii* tachyzoite uses a unique form of substrate-dependent motility to invade into and egress from cells of its host, to migrate across biological barriers, and to disseminate throughout the infected host. Motility is, therefore, central to the pathogenesis of toxoplasmosis (*Barragan et al., 2005*; *Harker et al., 2015*; *Meissner et al., 2002*). The parasite has no cilia or flagella and can move at speeds of up to 3 μm/s without the leading edge protrusions that drive the substrate-dependent motility of other eukaryotic cells. Within a three-dimensional (3D) extracellular matrix, the parasite moves along a complex helical trajectory characterized by regular oscillations in velocity and changes in trajectory curvature and torsion (*Leung et al., 2014*).

Motility is driven, at least in part, by a complex of proteins known as the glideosome, which is anchored to the inner membrane complex at the parasite periphery (*Figure 1A*). In the 'linear motor' model of motility, short actin filaments are bound through the bridging protein TgGAC to the cytosolic tails of transmembrane adhesins, such as TgMIC2, and translocated toward the posterior end of the parasite via a class XIV myosin, TgMyoA (*Frénal et al., 2017*; *Heintzelman, 2015*; *Jacot et al., 2016*; *Sibley, 2004*). If the extracellular domains of the transmembrane adhesins are attached to ligands on the substrate, the rearward translocation of actin results in forward movement of the parasite. Other apicomplexan parasites, including those that cause malaria (*Plasmodium spp.*) and cryptosporidiosis (*Cryptosporidium spp.*), express similar glideosome proteins, suggesting that the mechanism of motility is conserved (*Kappe et al., 1999*).

Although abundant evidence supports the importance of TgMyoA and other components of the glideosome in motility (*Meissner et al., 2002*; *Jacot et al., 2016*; *Andenmatten et al., 2013*; *Egarter et al., 2014*; *Tosetti et al., 2019*; *Dobrowolski and Sibley, 1996*; *Frénal et al., 2010*), several recent observations have raised questions about whether these proteins generate force as described by the linear motor model and, if so, whether this is the only motility mechanism available to the parasite (*Andenmatten et al., 2013*; *Gras et al., 2019*; *Meissner et al., 2013*; *Tardieux and Baum, 2016*; *Whitelaw et al., 2017*; *Rompikuntal et al., 2021*). Because parasite motility is essential for virulence, elucidation of the mechanism(s) underlying motility is of both fundamental cell biological interest and potential clinical relevance. One key piece of missing information needed for a full mechanistic understanding of motility is the directionality of the forces the parasite exerts on the external environment as it moves along its 3D helical trajectory. The linear motor model predicts continuous anterior-to-posterior forces along the entire periphery of the moving parasite. We show here using 3D traction force mapping that the predominant forces exerted by the parasite are instead organized into a highly localized circumferential ring of inward-directed force at a fixed location within the matrix. These inward-directed forces create a visible constriction of the parasite's plasma membrane, reflecting a circular zone of tight attachment of the parasite to the matrix through which the parasite passes as it progresses forward along its helical trajectory. In the absence of this circular attachment zone, the parasite is still able to move but this movement loses much of its directionality.

## Results

### Development of a quantitative 3D traction force mapping assay

As a first step toward visualizing the forces exerted by parasites moving in 3D, we used a bead displacement assay in the well-established Matrigel extracellular matrix model (*Leung et al., 2014*; *Whitelaw et al., 2017*; *Gras et al., 2017*). Fluorescent microspheres embedded within the Matrigel were pulled towards a moving parasite from nearby anterior, posterior, and lateral locations within the matrix, and these microspheres returned to their original positions after the parasite had passed

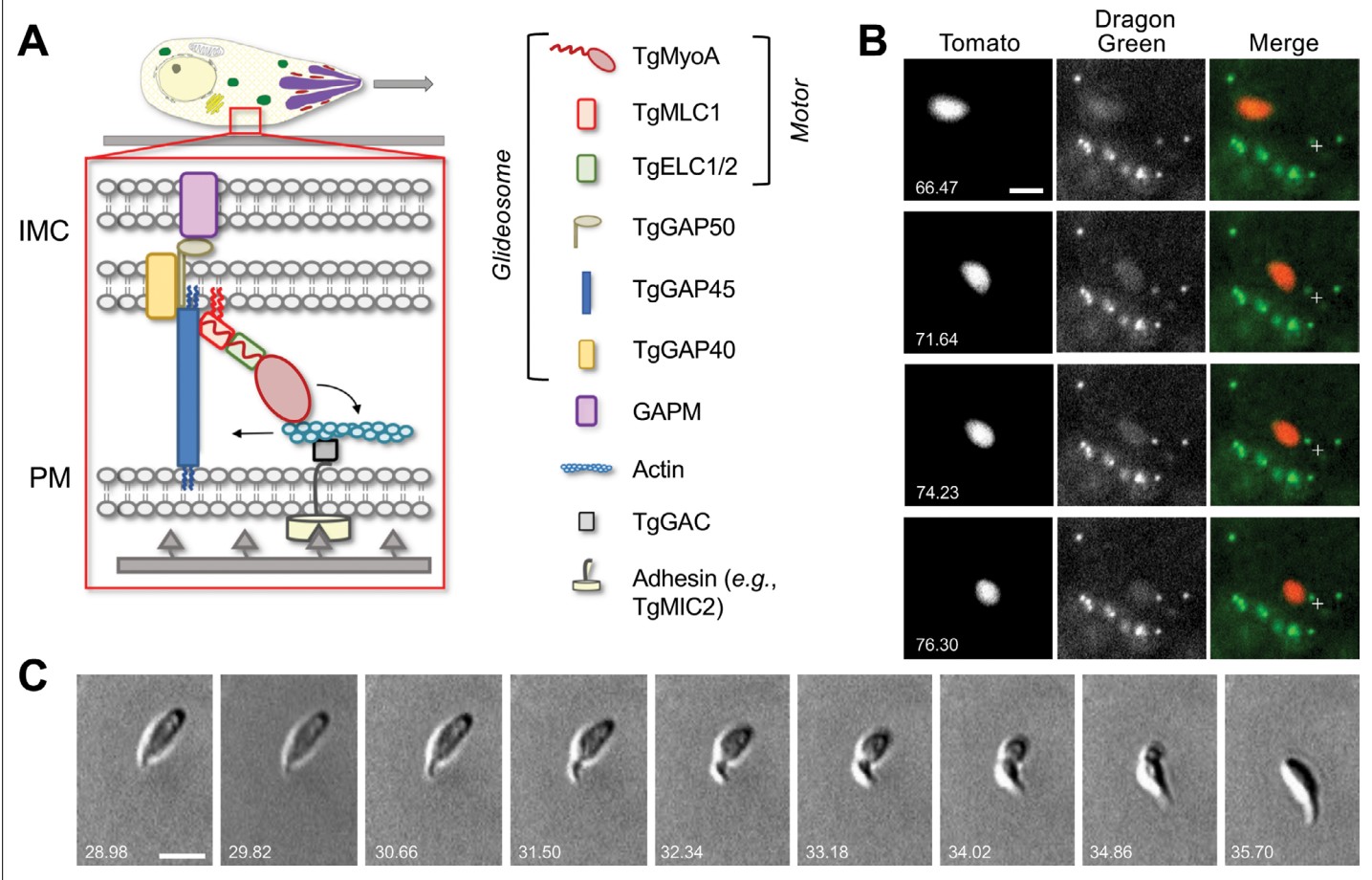

**Figure 1.** Parasites moving in 3D deform the surrounding Matrigel and undergo periodic constrictions. (**A**) In the linear motor model of motility, the TgMyoA motor (TgMyoA and its associated light chains, TgMLC1 and either TgELC1 or TgELC2) is anchored to the parasite's inner membrane complex (IMC) via TgGAP45 and the transmembrane proteins TgGAP40 and TgGAP50. The lumenal portion of GAP50 is thought to interact with GAPM, a protein that spans the inner IMC membrane and likely connects the entire glideosome to the underlying parasite cytoskeleton. Short actin filaments located between the parasite plasma membrane and the IMC are connected to ligands on the substrate through a linker protein, possibly TgGAC, which binds to the cytosolic tails of surface adhesins such as TgMIC2. The TgMyoA motor displaces the actin filaments rearward; because the motor is connected to the IMC and the actin is connected to the substrate, this causes the parasite to move forward relative to the substrate. The vertical blue and red squiggles on TgGAP45 and TgMLC1 denote lipid anchors that attach the proteins to the IMC (*Frénal et al., 2010*; *Rompikuntal et al., 2021*; *Foe et al., 2015*). Figure modified from *Rompikuntal et al., 2021* and re-used here under the terms of a Creative Commons Attribution 4.0 International license. (**B**) Sequential time series maximum intensity projections (in *z*) demonstrate that Dragon Green-labeled microspheres embedded within the Matrigel are displaced toward moving tdTomato-expressing parasites; note displacement of the green bead closest to the fixed crosshair in the merged images as the parasite moves from top left to bottom right. See *Video 1* for the entire time series, and *Video 2* for a second example of bead displacement by a moving parasite. (**C**) Brightfield images showing a constriction progressing from the anterior to the posterior end of a parasite as the parasite moves forward one body length; see *Video 3* for entire time series. All wild-type parasites analyzed (n=99) formed at least one constriction per body length of motility. Scale bars = 5 µm, timestamps in seconds.

(*Figure 1B*, *Videos 1 and 2*). These results demonstrate that the parasite does indeed exert a detectable force on the surrounding matrix. Unexpectedly, we also noted constrictions in the body of the fluorescent parasite as it moved (*Video 2*). These constrictions were even more apparent by brightfield microscopy (*Figure 1C*, *Video 3*). Each constriction formed at the apical end of the parasite and remained stationary relative to the matrix as the parasite moved through, reminiscent of the 'moving junction' through which the parasite penetrates during host cell invasion (discussed further below).

To develop a more quantitative force mapping assay and explore the relationship between the constrictions and matrix deformation, we used Alexa-Fluor 647-conjugated fibrinogen (*Owen et al., 2017*) to generate a fluorescent 3D fibrin matrix (*Figure 2A* and *Video 4*). The use of a fluorescent matrix enabled mapping of matrix displacements at all spatial points in our imaged volume rather than at the limited number of discrete positions offered by fluorescent microspheres. Parasite motility

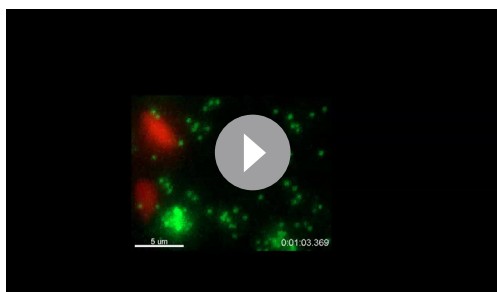

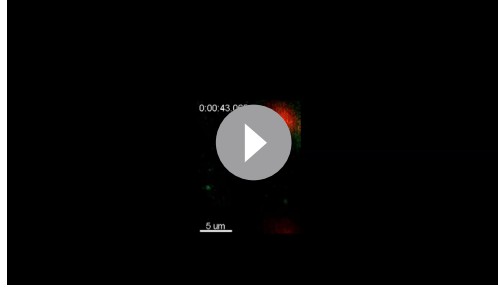

**Video 1.** Maximum intensity projection in *z* of microspheres (green) being displaced due to parasite (red) movement within the Matrigel. Scale bar = 5 μm, time is shown in hr:min:sec. Single frames from this video are shown in Figure 1B.

https://elifesciences.org/articles/85171/figures#video1

**Video 2.** Microsphere (green) being displaced towards a parasite (red) moving in Matrigel within in a single *z* plane. Scale bar = 5 μm, time is shown in hr:min:sec.

https://elifesciences.org/articles/85171/figures#video2

in matrices generated using 2.25 mg/ml fibrinogen was indistinguishable from that observed in Matrigel (*Figure 2B* and *Figure 2—figure supplement 1*). Visual comparison of successive image volumes confirmed that parasites moving through fluorescent fibrin also deform the matrix (*Figure 2C* and *Video 5*) and undergo constrictions (see below) similar to those observed in Matrigel.

To calculate and map the deformations of the fluorescent fibrin matrix, we used the Fast Iterative Digital Volume Correlation (FIDVC) algorithm developed by Christian Franck's group (*Patel et al., 2018*) to iteratively determine the 3D fibrin displacement fields between consecutive volumetric image stacks in our time series (*BarKochba et al., 2015*). Each image volume is divided into 16,807 subvolumes (49 x 49 x 7 subvolumes in *x*, *y*, *z*). The algorithm compares the voxel intensity pattern between corresponding subvolumes at two consecutive time points, calculating one 3D displacement vector per subvolume. The data can be displayed as a map for the entire imaging volume or for any 2D plane (*x-y*, *x-z*, or *y-z*; see *Video 6*). Most of the 2D images below show the displacement maps in the *x-y* plane on one of the seven *z* subvolume levels.

To filter out background noise and identify displacements caused by the parasite, we calculated displacement vectors during periods when no parasites were moving within our imaging volume and used these to establish a displacement detection threshold for each dataset (see *Figure 3* and Methods for details). Using this approach, the limit of detection for 3D displacement magnitudes in the system ranges from 42 to 46 nm; a similar 3D displacement detection threshold (43–46 nm) was observed in samples containing no parasites (see Figure 6E below), validating the approach. When analyzing data in the *x-y* plane only, the background cutoff was calculated using the *x-y* displacement magnitudes (rather than the *x-y-z* displacement magnitudes); this lowered the displacement detection threshold to 28–31 nm.

In order to translate matrix displacements into the magnitude of forces generated by the parasite, we determined the viscoelasticity of the fibrin matrix using laser trapping (see Methods). The fibrin gel behaves as a predominantly elastic matrix (*Figure 3—figure supplement 1*) with an elastic modulus of 15.6 pN/μm and a viscous modulus of 0.09 pN*s/μm. The minimal detectable force magnitude in this system is therefore 0.66–0.72 pN in 3D and 0.45–0.49 pN in the *x-y* plane.

## Parasite constrictions are tightly linked to periodic bursts of matrix deformation and motility

With this system for 3D traction force mapping in hand, we analyzed the pattern and directionality

**Video 3.** Brightfield imaging of a parasite undergoing a single constriction. Scale bar = 5 μm, time is shown in hr:min:sec. Single frames from this video are shown in Figure 1C.

https://elifesciences.org/articles/85171/figures#video3

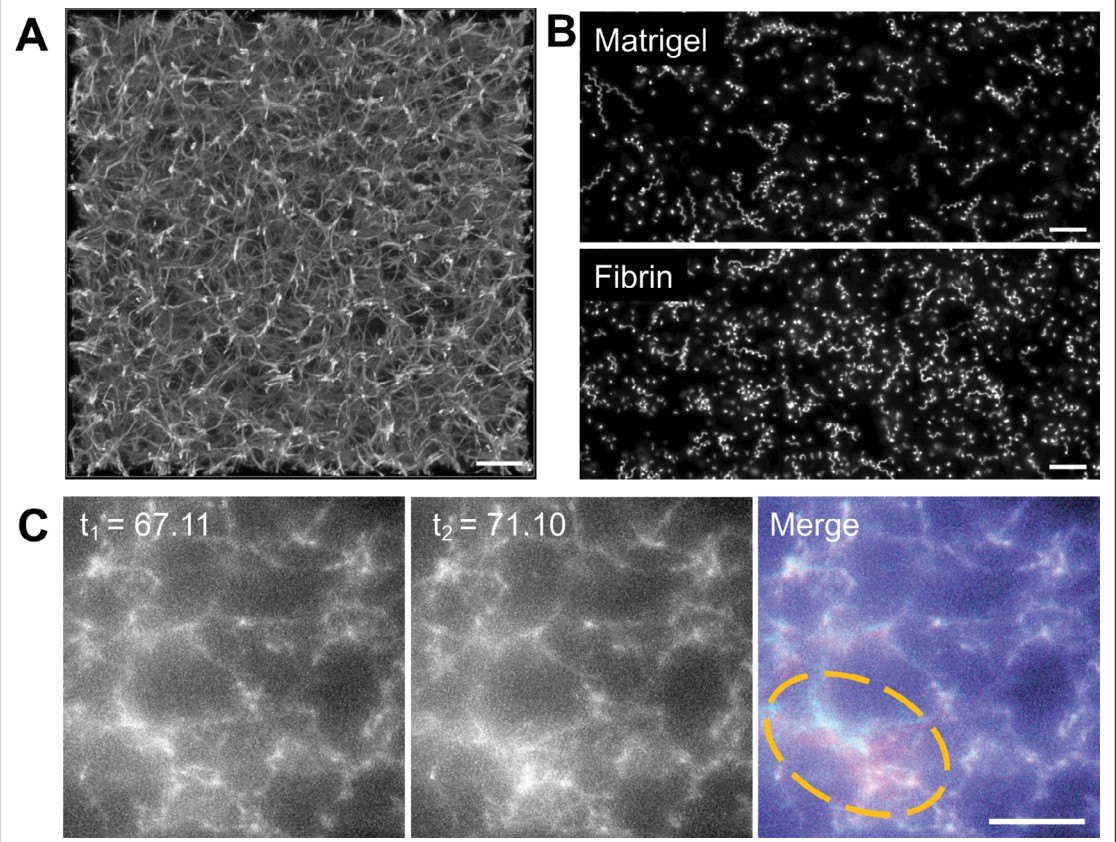

**Figure 2.** Parasite motility within a 3D fibrin matrix. (**A**) Confocal imaging of a 2.25 mg/ml fluorescent fibrin gel. A maximum fluorescence intensity projection of 51 *z*-slices captured 0.25 µm apart is shown; scale bar = 10 µm. (**B**) Maximum fluorescence intensity projections showing the trajectories of tdTomato-expressing wildtype parasites moving in 3D in Matrigel (top) *vs.* fibrin (bottom). See *Figure 2—figure supplement 1* for quantitative comparison of the motility parameters in the two matrices. Scale bar = 40 µm. (**C**) A parasite moving in a 1.3 mg/ml fluorescent fibrin matrix (see *Video 5*) visibly deforms the matrix, as evident from the non-coincident fluorescence signals in the highlighted area at the time points indicated (merge). Scale bar = 10 µm, timestamps in seconds.

The online version of this article includes the following figure supplement(s) for figure 2:

**Figure supplement 1.** Quantitative comparison of parasite motility in fluorescent fibrin *vs.* Matrigel.

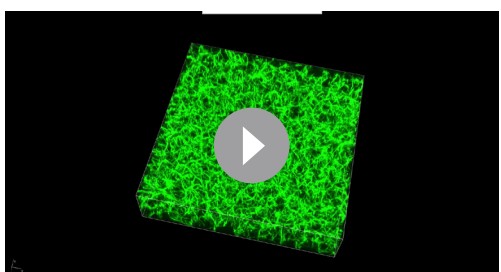

**Video 4.** Confocal fluorescence imaging of a 2.25 mg/ml fluorescent fibrin gel. 51 *x-y* slices captured 0.25 µm apart in *z* were reassembled into the volumetric view shown. Dimensions of the imaged volume are shown at bottom left. See Figure 2A for maximum intensity projection of these data.

https://elifesciences.org/articles/85171/figures#video4

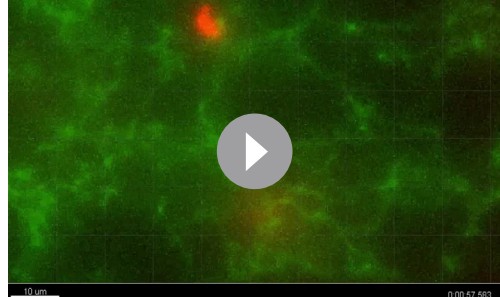

**Video 5.** Deformation of fluorescent fibrin (green) by a parasite (red; bottom) as it moves up through the image stack. Scale bar = 10 µm, time is shown in hr:min:s. Single frames from this video are shown in Figure 2C.

https://elifesciences.org/articles/85171/figures#video5

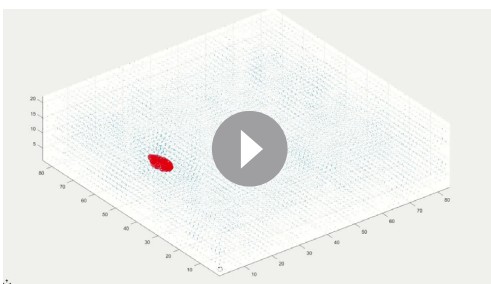

**Video 6.** 3D displacement map surrounding the moving parasite shown in the second panel of Figure 4C and frame 2 of *Video 7*. Length of arrows indicating displacement magnitude are multiplied 10-fold for display. The parasite is outlined on each z plane with red circles. Single frames from this video displaying projections onto the *x-y, x-z* and *y-z* planes are shown in Figure 4—figure supplement 1.
https://elifesciences.org/articles/85171/figures#video6

of forces produced by wild-type parasites as they moved through the fibrin matrix. Plotting fibrin displacement as *x-y* maps showed that the matrix deforms in toward the parasite from all *x-y* directions at discrete, periodic time points along the trajectory, before relaxing back to its initial position (*e.g.*, *Figure 4A–D* and *Videos 7 and 8*). Viewing the data as projections along each axis demonstrates that the matrix is simultaneously pulled in toward the parasite in all three dimensions (*Video 6* and *Figure 4—figure supplement 1*). In approximately 50% of cases, inward deformation occurred between two consecutive image volumes (1.6 seconds apart) and was followed in the next image volume by relaxation (*e.g.*, *Figure 4A–C*). In most other cases, the inward deformation was held through multiple time points before relaxing (*Figure 4—figure supplement 2*). During periods of motility when the matrix was not being pulled inward or relaxing outward, little deformation of the matrix above background was detectable (e.g. *Figure 4C*, leftmost and rightmost panels). This observation suggests either that the parasite pulls on the matrix only during some portions of its trajectory or that the parasite produces both strong and weak deformations, and the weak deformations – which could be continuous – are below our level of detection.

When the displacement maps were overlaid onto the images of the moving parasites, it was immediately apparent that the timing of each large matrix deformation coincided with the presence of a parasite constriction and that the displacement vectors pointed primarily in towards the constriction (*Figure 4C and D*; additional examples are shown in *Figure 4—figure supplement 3*). The fact that the displacements were directed inwards rather than away from the parasite indicates that the pinching of the parasite membrane results from pulling forces generated by the parasite rather than the parasite squeezing through a narrow pore within the matrix, which would push the matrix away. When a constriction persisted over multiple time points, the fibrin continued to deform into the constriction and/or to hold the deformation over those same time points as the parasite passed through (e.g. *Figure 4D*).

The time it took a parasite to move completely through a constriction ranged from 1.6 to 56.2 seconds (average 6.8±5.2 seconds, measured in brightfield; n=99 parasites, 188 constrictions) with the longer times due to a stall in forward progression of a constricting parasite. The time between one constriction finishing and another beginning was also variable (*Figure 5A*). In 46% of the cases, the end of one constriction was followed immediately by the beginning of another, while in 14% of the cases a second constriction would start before the first constriction finished (*Figure 5A and B* and *Video 9*). The speed of the parasite's forward motion changed as it moved through the constriction, increasing once the constriction passed the halfway point on the parasite's longitudinal axis (*Figure 5—figure supplement 1*; n=20 parasites, 24 constrictions). Most importantly, all parasites that move at least one body length undergo a constriction and the total trajectory length is directly proportional to the number of constrictions observed (*Figure 5C*), strongly suggesting that the parasite-generated forces that create the constriction play an important role in forward movement.

The much reduced width of a constricting parasite (2.6±0.4 μm *vs.* 1.4±0.2 μm at the midway point along the parasite's longitudinal axis before and during a constriction, respectively; *Figure 5D*, n=15 parasites, 19 constrictions) led us to question whether the constriction presents a physical obstacle to the forward motion of the nucleus (normal nuclear diameter measured by DNA staining = 2.0 ± 0.3 μm). In parasites labeled with both a fluorescent DNA stain and a fluorescently labeled antibody against the major surface protein, TgSAG1 (to visualize the constriction), the normally round parasite nucleus did indeed become thinner and more elongate as it passed through the constriction and

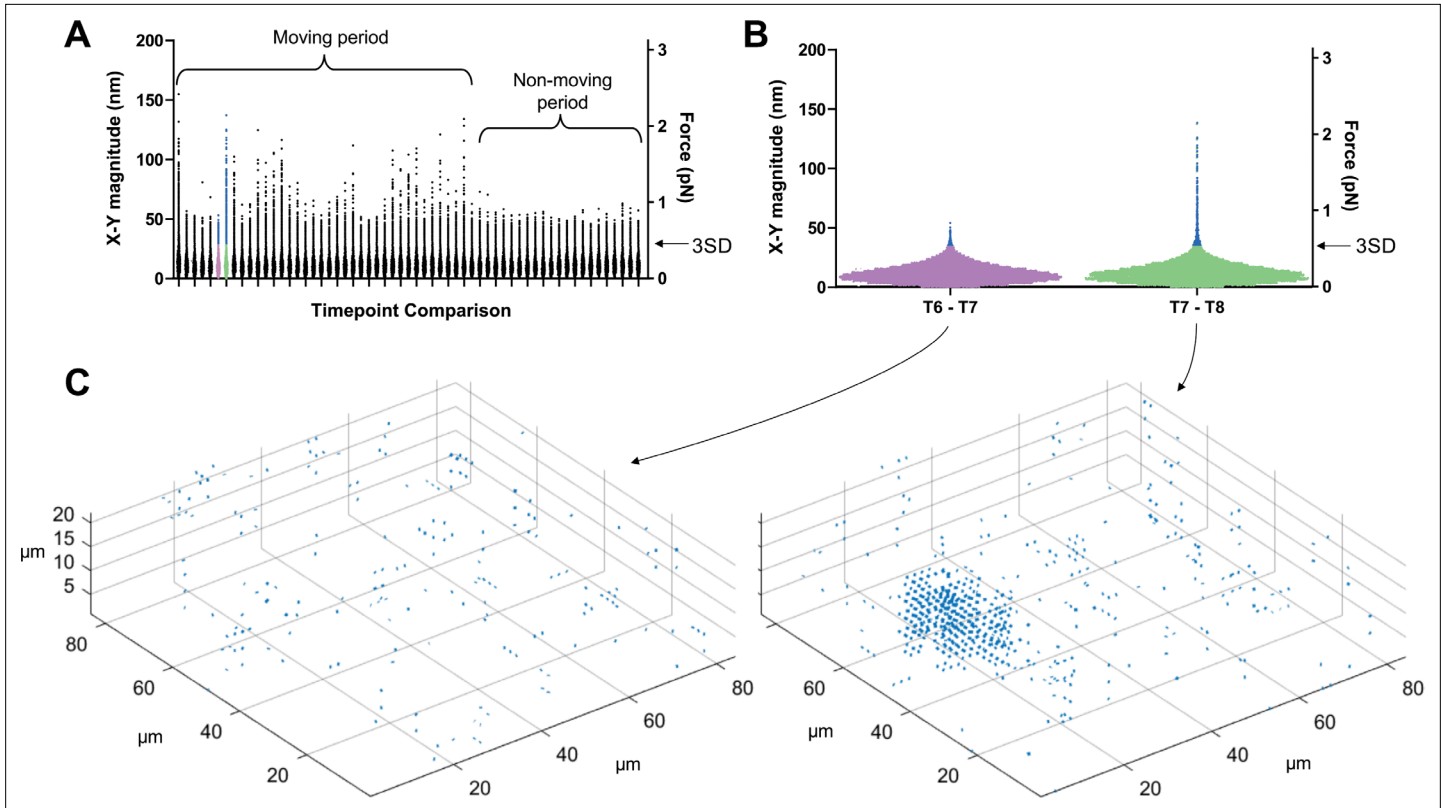

**Figure 3.** Determining threshold of detection in 3D traction force mapping. (**A**) A fluorescent fibrin matrix containing wild-type parasites was imaged over 96 seconds (60 successive image volumes). FIDVC was then used to calculate the 16,807 3D fibrin displacement vectors between pairs of successive time points. The magnitudes of the 16,807 *x-y* displacement vectors are plotted for each of the 59 pairwise time point comparisons. This dataset contained a single moving parasite, which moved during the first part of the time course but not at the end, as indicated. For each time course, the two consecutive time points that gave the lowest mean *x-y* displacement vector magnitude were used to set the background threshold for that dataset: any displacements less than three standard deviations (3SD) above this mean were considered background noise. (**B**) Expanded view of the 16,807 datapoints from the two time point comparisons shown in purple and green in panel A. Arrow indicates the 3SD background threshold; all points above this threshold are colored blue in panels A and B. The moving parasite caused little deformation of the matrix above background between time points 6 and 7 and a greater amount of deformation between time points 7 and 8. (**C**) Displacement vectors with magnitudes greater than the 3SD background threshold for the two highlighted time point comparisons were mapped back onto the imaging volume; the cluster of displacement vectors evident in the time points 7–8 comparison corresponds to the position of the single moving parasite in the volume.

The online version of this article includes the following figure supplement(s) for figure 3:

**Figure supplement 1.** Rheological properties of the fibrin matrix.

regained its round shape once it was through (*Figure 5E*). The ratio of the nuclear dimensions perpendicular and parallel to the long axis of the parasite was 0.93±0.08 in a non-constricting parasite and 0.50±0.06 as the nucleus passed through a constriction (n=9 parasites, 21 constrictions).

Interestingly, the distribution of the fluorescent anti-TgSAG1 antibody on the parasite surface changed as the parasite moved: much of the fluorescent signal appeared to be 'swept' from anterior to posterior at the constriction as the parasite moved through (*Figure 5—figure supplement 2* and *Video 10*). The redistributed antibody was ultimately shed and deposited behind the moving parasite in the form of a fluorescent helical trail within the Matrigel (*Figure 5—figure supplement 2* and *Video 10*).

## Parasites that do not generate constrictions do not detectably deform the matrix

Since TgMyoA is thought to play a central role in force production during parasite motility (*Figure 1A*), we conducted force mapping experiments using a parasite line lacking TgMyoA (*Egarter et al., 2014*). As previously reported (*Whitelaw et al., 2017*), these parasites are much less motile in 3D

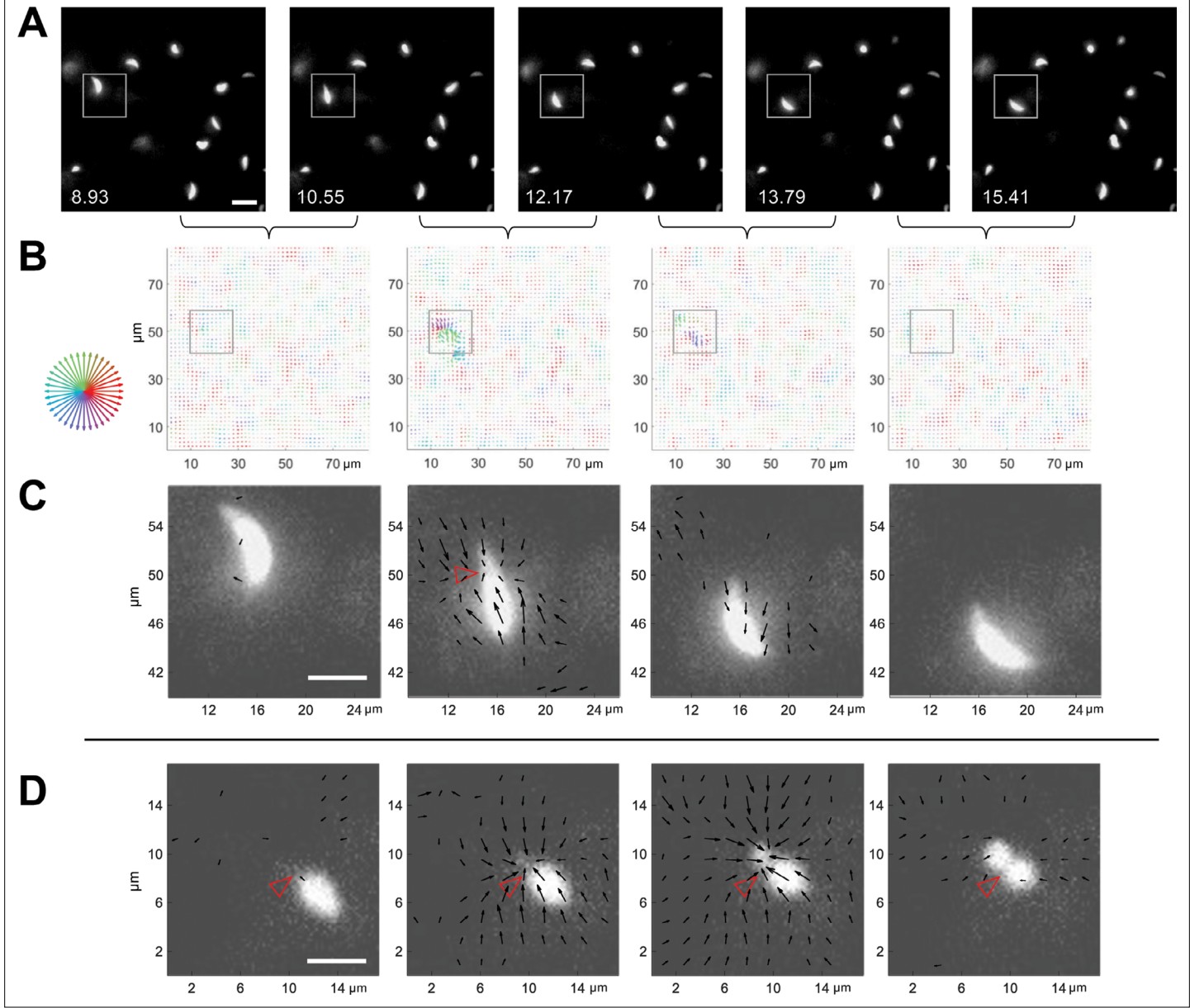

**Figure 4.** 3D traction force mapping in fluorescent fibrin reveals that the matrix is periodically pulled in toward the constriction during parasite motility. (**A**) Sequential time series images, in a single *z*-plane, of a tdTomato-expressing parasite moving in fibrin (boxed). Scale bar = 10 µm, timestamps in seconds. (**B**) Force maps from the corresponding *z* subvolume plane of the data shown in panel A; each map depicts the motions of the fluorescent fibrin matrix between the two consecutive time points, as indicated by the brackets. Arrow length (multiplied 24-fold for display) indicates the magnitude of matrix displacement and arrow color the directionality (see color wheel). See *Video 7* for the entire time series. (**C**) Zoomed images showing the force maps from the boxed region of panel B, after background subtraction (see text), overlaid on the parasite images from panel A. Length of arrows indicating displacement magnitude are multiplied 15-fold for display. Note inward displacement of the matrix in the second overlay image, outward displacement (relaxation) of the matrix in the third image, and no detectable displacement vectors in the fourth image even though the parasite continues to move. (**D**) Zoomed overlays of the force maps (background subtracted) and images of a moving parasite from a second dataset. The inwardly displaced matrix in this example did not relax back to its original position within the time frame of the experiment. Arrow lengths indicating displacement magnitude are multiplied 15-fold for display. See *Video 8* for the entire time series. Scale bars in panels C and D=5 µm. Empty red arrowheads indicate position of the constriction.

The online version of this article includes the following figure supplement(s) for figure 4:

**Figure supplement 1.** All displacement vectors point in towards a moving parasite.

**Figure supplement 2.** Pattern of pulling, holding, and release of the matrix during individual constriction events.

**Figure supplement 3.** Additional examples showing that matrix displacement is directed primarily in towards the constriction in moving parasites.

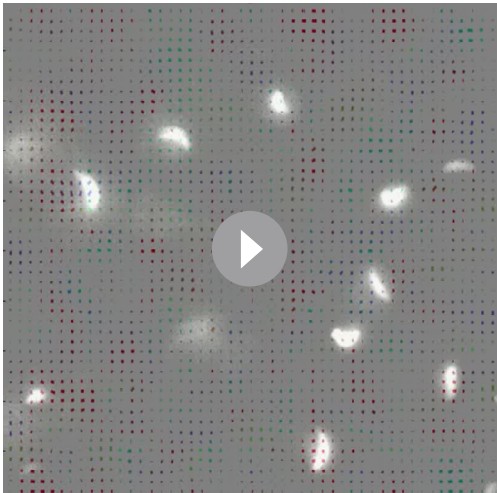

**Video 7.** x-y displacement map overlaid on the moving parasite shown in Figure 4A–C. Arrow size corresponds to relative displacement magnitude and arrow color to displacement direction as described in Figure 4.
https://elifesciences.org/articles/85171/figures#video7

than wild-type parasites: only 18 of 769 TgMyoA knockout parasites examined (2%) moved one body length or more during a 302-second time course, compared to 99 of 278 wild-type parasites (36%). Of the 18 TgMyoA knockout parasites that moved at least one body length (e.g. *Figure 6A*), none showed clear constrictions at a fixed point within the matrix like those seen in wild-type parasites. The pattern of movement was also different from wild-type parasites; rather than moving along smooth helical trajectories, the TgMyoA knockout parasites moved in tightly twisting arcs or in a stair-like pattern consisting of right-angled turns associated with a sharp bend in the body of the parasite (*Figure 6A* and *Video 11*). In stark contrast to the wild-type parasites, we also saw no evidence by traction force mapping of a ring of inward-directed forces produced by the TgMyoA knockout parasites. In fact, we did not detect any displacement of the matrix above background noise by these parasites (compare *Figure 6B* to *Figure 6D and E*; see also *Figure 6—figure supplement 1*, n=16 parasites), suggesting that any forces produced by the moving knockout parasites are below the limit of detection in our traction force assay.

We also tested whether the surface adhesin, TgMIC2, plays a role in generating the constriction-associated inward forces on the matrix. The motility defect in parasites lacking TgMIC2 (*Gras et al., 2017*) is less severe than the defect in parasites lacking TgMyoA, with 36 of 297 TgMIC2 knockout parasites (12%) capable of moving more than one body length (see also *Gras et al., 2017*; *Huynh and Carruthers, 2006*). However, these parasites often moved in an erratic, stop and start pattern, bending sharply and moving in tightly twisting arcs (*Figure 6A* and *Video 12*). The TgMIC2 knockout parasites also failed to form constrictions and did not detectably deform the fibrin matrix (*Figure 6C*, n=36 parasites, *Figure 6—figure supplement 2*, n=18 parasites) despite being able to move at maximum speeds similar to wild-type parasites (*Gras et al., 2017*). Taken together, these data confirm that motility is significantly altered, but not completely eliminated, in parasites lacking TgMyoA or TgMIC2. Furthermore, the data demonstrate that both proteins are required to produce the constrictions and the constriction-associated inward forces on the matrix observed during the motility of wild-type parasites.

## TgMIC2-deficient parasites move less directionally

The larger number of TgMIC2 knockout parasites capable of long runs of motility (compared to the TgMyoA knockouts) enabled us to analyze their phenotype in greater detail. The nuclei of parasites lacking TgMIC2 did not change shape as they moved in 3D (*Figure 7A* and *Figure 7—figure supplement 1A*), providing further evidence that these parasites do not undergo the constrictions seen in wild-type parasites. Given their apparent propensity for tight turns, we also quantified the ability of the TgMIC2 knockout parasites to move

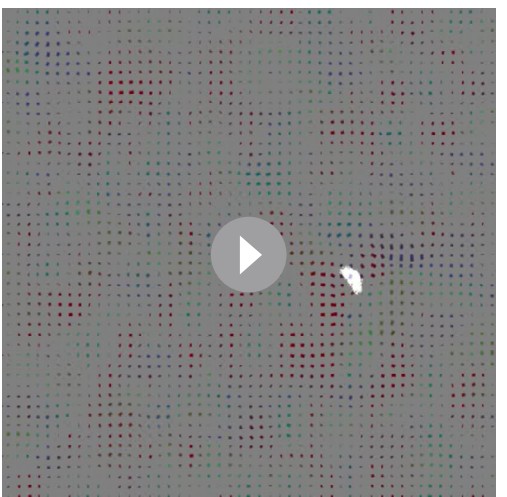

**Video 8.** x-y displacement map overlaid on the moving parasite shown in Figure 4D. Arrow size corresponds to relative displacement magnitude and arrow color to displacement direction as described in Figure 4.
https://elifesciences.org/articles/85171/figures#video8

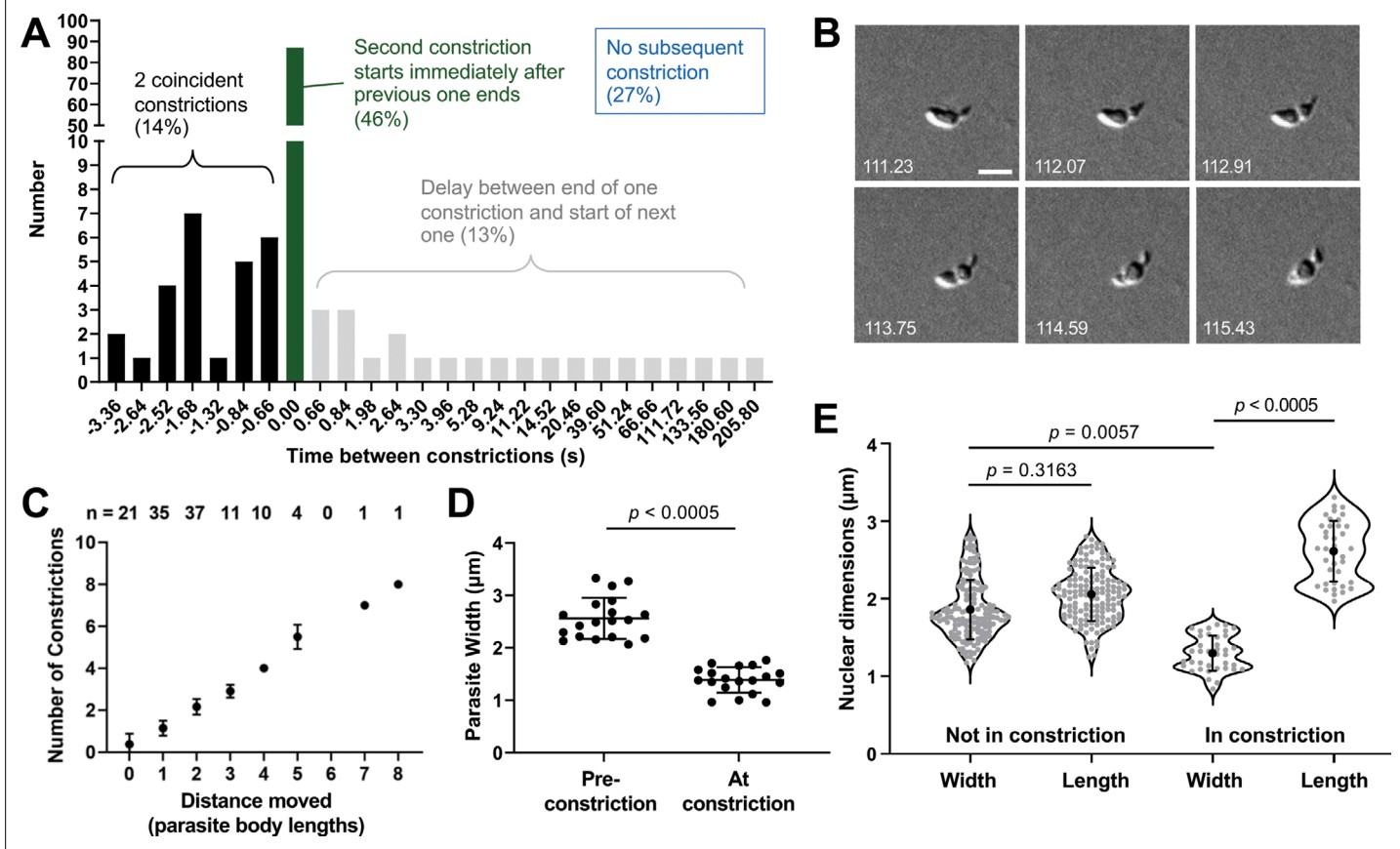

**Figure 5.** The constrictions are tightly linked with motility and are sufficiently narrow to deform the parasite nucleus. (**A**) The time interval between constrictions during motility of individual parasites (n=99 parasites, 188 constrictions). A negative time interval (black bars) corresponds to the presence of two constrictions in the same parasite for the indicated amount of time. Green bars denote back-to-back constrictions and grey bars denote a delay between constrictions for the indicated amounts of time. For 27% of the constrictions (not plotted), no subsequent constriction occurred within the time frame of the experiment. (**B**) Example of a parasite with two concurrent constrictions; see *Video 9* for entire time series. Brightfield images; scale bar = 5 µm; timestamps in seconds. (**C**) Number of constrictions observed *vs.* distance traveled, expressed in terms of binned parasite body lengths (i.e. multiples of 7 µm). (**D**) Width of individual moving parasites, measured at the halfway point along their longitudinal axis, either before the halfway point reached the constriction ('Pre-constriction') or as the halfway point was passing through the constriction ('At constriction'). Data from two independent biological replicates were combined; n=15 parasites, 19 constrictions. (**E**) Dimensions of the nucleus (determined by Hoechst 33342 staining) in individual parasites during time points when the nucleus was located either ahead of or behind the constriction ('Not in constriction') *vs.* passing through the constriction ('In constriction'). Nuclear length and width are defined as the nuclear diameters parallel and perpendicular to the long axis of the parasite, respectively. Data from two independent biological replicates were combined; n=9 parasites, 21 constrictions. In panels D and E, bars indicate mean (± SD); samples were compared by Student's two-tailed t-test.

The online version of this article includes the following figure supplement(s) for figure 5:

**Figure supplement 1.** Parasites move faster after the constriction has past the midway point on the parasite's longitudinal axis.

**Figure supplement 2.** Fluorescent antibody against TgSAG1 is depleted from the parasite surface anterior to the constriction and shed into the Matrigel matrix.

directionally, by comparing displacement distance (first to last point) to total trajectory length. While parasites with and without TgMIC2 traveled along trajectories of similar mean length (38±9 µm vs. 42±8 µm, respectively), parasites lacking TgMIC2 moved approximately half as far from their starting point as wild-type parasites (*Figure 7B* and *Figure 7—figure supplement 1B*). These data suggest that the circumferential zone of attachment to the matrix, which requires TgMIC2, functions to convert meandering motility into more linear, directed forward progression.

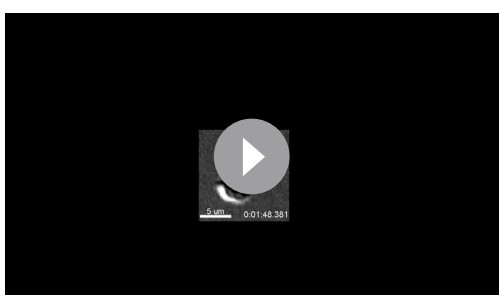

**Video 9.** Brightfield imaging of a parasite undergoing two constrictions at the same time. Scale bar = 5 μm, time is shown in hr:min:s. Single frames from time points 0:1:51.23 – 0:1:55.43 of this video are shown in Figure 5B.
https://elifesciences.org/articles/85171/figures#video9

## Discussion

The linear motor model of apicomplexan motility predicts that a parasite moves by pulling continuously on the matrix in an anterior-to-posterior direction along the length of the parasite. It was, therefore, unexpected that the predominant forces revealed by our 3D traction force mapping assay were instead periodic and highly localized, pointing inwards from all directions towards a circumferential location on the parasite surface. This annular band of parasite-generated, inward-directed force forms at a fixed position within the matrix and creates a visible constriction at the parasite periphery. The constriction passes from the anterior to the posterior end of the parasite as the parasite moves forward. Typically, when the constriction reaches the parasite's posterior end a new constriction forms at the apical end and the cycle repeats. These results suggest that the moving parasite periodically assembles ring-shaped zones of tight attachment to the matrix, which deform the matrix and serve as stationary platforms that the parasite propels itself through, one body length at a time.

### Attachment to the substrate, force generation, and motility in 2D *vs.* 3D

Both malaria sporozoites and *T. gondii* tachyzoites are capable of moving in circles on 2D surfaces. The pioneering work of Munter, Frischknecht and colleagues combined traction force mapping with reflection interference contrast microscopy (RICM) during the 2D circular gliding of malaria sporozoites (*Münter et al., 2009*). They concluded that the parasite adheres to the substrate at its apical and posterior ends; each of these adhesion sites produces stalling forces parallel to the long axis of the parasite, and disengagement of the adhesion sites drives forward motion. Traction forces perpendicular to the direction of movement were also detected at the center of the parasite but these were interpreted as nonspecific and nonproductive. The apical and posterior adhesion sites were shown to move forward *with* the parasite, rather than remaining stationary with respect to the substrate.

*T. gondii* tachyzoites demonstrate a second form of productive motility on 2D surfaces called helical gliding: the banana-shaped parasite starts with its left side in contact with the substrate (where the convex face is defined as dorsal), moves forward one body length while rotating along its longitudinal axis until it lies on its right side, and then flips back to its left side to repeat the cycle (*Håkansson et al., 1999*). This form of motility is the equivalent of the fully helical trajectories seen in 3D but constrained in 2D by contact of the curved parasite with the rigid substrate. In RICM studies of 2D helical gliding (*Tosetti et al., 2019*; *Pavlou et al., 2020*) the tachyzoite was seen to first attach to the substrate at its apical end. The zone of attachment then expanded along the length of the parasite and, as the parasite moved forward and its apical end twisted up off the cover slip, the zone of attachment shrunk until it was reduced to a spot at the posterior end. In contrast to *Plasmodium* sporozoites, the position of the tachyzoite adhesion site remained fixed relative to the substrate and the parasite slid over this attachment site as the parasite moved forward. 2D traction force mapping revealed forces parallel, but not perpendicular, to the longitudinal axis of the

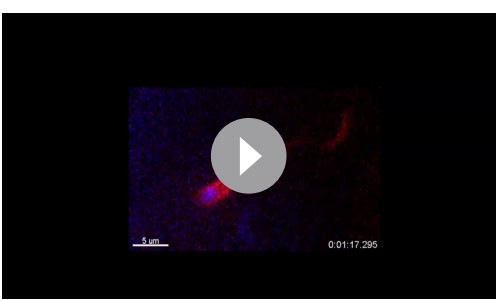

**Video 10.** Parasite labeled with Alexa546-conjugated anti-TgSAG1 (red) and Hoechst 33342 (blue) undergoing a constriction. Scale bar = 5 μm, time is shown in hr:min:s. Note the helical trail of shed fluorescent antibody behind the moving parasite. Single frames from this video are shown in Figure 5—figure supplement 2.
https://elifesciences.org/articles/85171/figures#video10

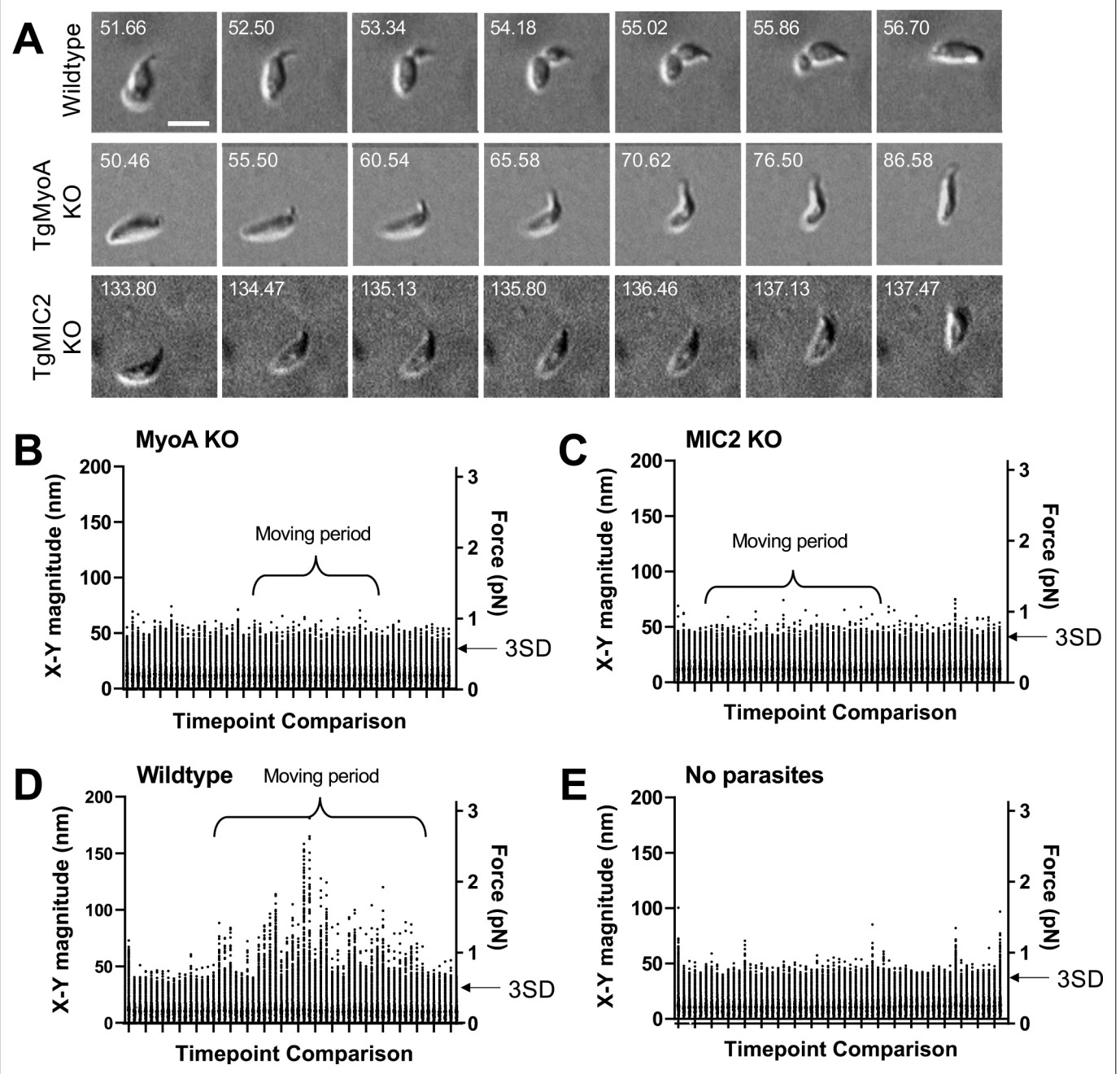

**Figure 6.** 3D motility and force mapping of parasites lacking TgMyoA or TgMIC2. (**A**) Brightfield images showing TgMyoA knockout and TgMIC2 knockout parasites moving within a Matrigel matrix without a detectable constriction. A wild-type parasite undergoing a typical constriction is shown for comparison. The TgMyoA knockout parasite makes right-angled 'stairstep' turn, and the TgMIC2 knockout parasite moves erratically in a tightly twisting arc. Scale bar = 5 μm, timestamps in seconds. See *Videos 11 and 12* for the entire time series for the TgMyoA and TgMIC2 knockout parasites. (**B–E**) As in *Figure 3A*, the plots show the magnitudes of the 16,807 fibrin *x-y* displacement vectors between each pair of successive image volumes generated in experiments using: (**B**) TgMyoA knockout parasites; (**C**) TgMIC2 knockout parasites; (**D**) wild-type parasites; and (**E**) no added parasites (fibrin only). Periods during the 96-second time course when parasites were moving are indicated. For each time course, the two consecutive time points that gave the lowest mean displacement magnitude were used to set the background threshold: any displacements less than three standard deviations (3SD) above this mean were considered noise for that dataset. Examples of the force maps surrounding motile TgMyoA and TgMIC2 knockout parasites are shown in *Figure 6—figure supplements 1 and 2*, respectively.

The online version of this article includes the following figure supplement(s) for figure 6:

**Figure supplement 1.** The small number of TgMyoA knockout parasites that move more than one body length produce no detectable force on the fibrin matrix.

**Figure supplement 2.** Moving TgMIC2 knockout parasites produce no detectable force on the fibrin matrix.

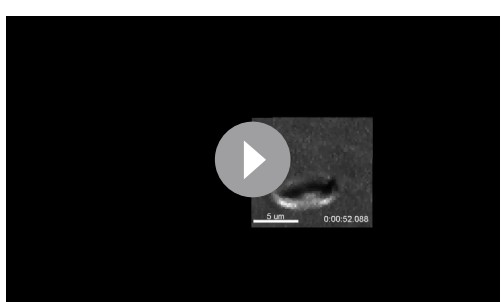

**Video 11.** Brightfield imaging of a moving TgMyoA knockout parasite. Scale bar = 5 μm, time is shown in hr:min:sec. Single frames from this video are shown in Figure 6A.

https://elifesciences.org/articles/85171/figures#video11

tachyzoite; these traction forces, together with spring-like forces generated by the parasite's microtubule cytoskeleton, were postulated to produce forward motion (*Pavlou et al., 2020*).

When these previous studies of *T. gondii* 2D motility are combined with the 3D results presented here, a consensus emerges that the basic unit of motility consists of apical attachment to the substrate, followed by apical-to-posterior translocation of the adhesion site, which is fixed in its position relative to the substrate. When the adhesion site reaches the posterior end of the parasite, a new apical attachment site is formed, and the cycle begins again. The adhesion site in a 3D matrix is circular, and the pulling forces exerted by the parasite on the matrix manifest as a visible constriction in the parasite's plasma membrane. In 2D, no deformation of the parasite has been reported at the adhesion site, likely reflecting differences in attachment to the substrate via a point/patch on the parasite surface *vs.* circumferential attachment. We also saw no evidence in 3D for a persistent stall force at the rear of the parasite. The biggest difference between the 2D and 3D force mapping data is that, in 3D, all force vectors point towards one highly localized circumferential region on the surface of the parasite, and many of these force vectors contained a strong component perpendicular to the long axis of the parasite.

*Munera Lopez et al., 2022* also recently noted that tachyzoites undergo constrictions as they move through Matrigel. These authors ascribed the constrictions to the parasite having to squeeze through small pores in the matrix. Our force mapping results and the regularity with which the constrictions form at the apical end of the parasite argue strongly that the constrictions do not result from the parasite pushing its way through pores, but rather from parasite-generated pulling forces on the matrix. Annular constrictions that remain stationary relative to the surrounding environment have also previously been reported along the length of *Plasmodium* ookinetes as they move through peritrophic membrane and microvillar network of the mosquito midgut (*Freyvogel, 1966*; *Vlachou et al., 2004*; *Zieler and Dvorak, 2000*), suggesting that parasite-generated circular zones of attachment to the environment may be a conserved feature of apicomplexan motility in 3D.

## 3D motility and invasion: variations on a theme

During the invasion of host cells by *T. gondii* and other apicomplexan parasites, proteins secreted from the parasite's apical organelles assemble a ring-shaped zone of tight attachment between the membranes of the two cells. This 'moving junction' does not in fact move: it is anchored to the host cytoskeleton and is hypothesized to provide a fixed platform against which the parasite exerts force (*Alexander et al., 2005*; *Besteiro et al., 2011*; *Dubremetz, 1998*; *Lebrun et al., 2005*; *Porchet-Hennere and Torpier, 1983*; *Aikawa et al., 1978*). Parasite surface adhesins, which are engaged with the junction via their extracellular domains, are thought to be translocated by MyoA in an anterior-to-posterior direction, pushing the parasite through the junction and into the host cell. The body of the parasite narrows dramatically as it passes through the moving junction, bearing a striking resemblance to the constrictions we report here during 3D motility.

Two central players in moving junction formation in *T. gondii* are TgAMA1, an adhesin secreted onto the parasite surface from the micronemes, and TgRON2, which is secreted by the parasites into the host cell plasma membrane and serves

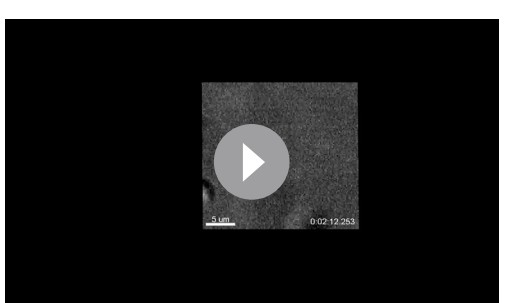

**Video 12.** Brightfield imaging of a moving TgMIC2 knockout parasite. Scale bar = 5 μm, time is shown in hr:min:s. Single frames from this video are shown in Figure 6A.

https://elifesciences.org/articles/85171/figures#video12

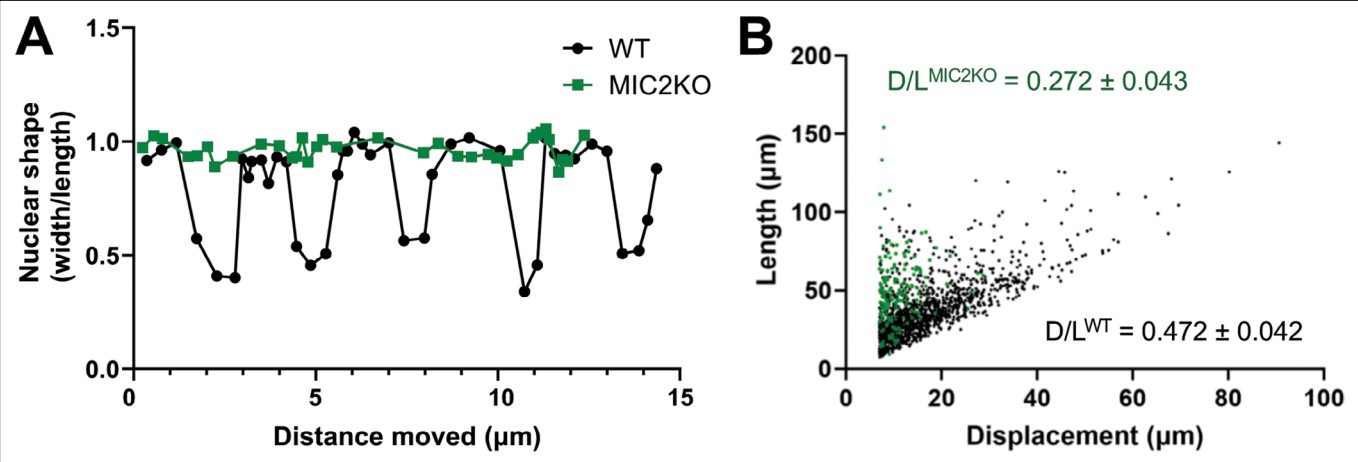

**Figure 7.** Knockout of TgMIC2 results in the loss of the constriction and less directional motility. (**A**) Representative plots of nuclear shape (ratio of the nuclear diameters perpendicular *vs.* parallel to the long axis of the parasite) in one wildtype and one TgMIC2 knockout parasite (black circles and green squares, respectively), as the parasites moved over time. The wild-type parasite underwent multiple concurrent constrictions along its trajectory. For the combined data from multiple parasites, see *Figure 7—figure supplement 1A*. (**B**) Ratio of parasite displacement (D) to trajectory length (L) for wildtype (n=1399) and TgMIC2 knockout (n=165) parasites (black circles and green squares, respectively). The calculated means (± SD) are from three independent biological replicates, each consisting of three technical replicates (see *Figure 7—figure supplement 1B*). In panels A and B, 'WT' refers to the TgMIC2 LoxP parasites (*Gras et al., 2017*) before treatment with rapamycin to excise *MIC2*.

The online version of this article includes the following figure supplement(s) for figure 7:

**Figure supplement 1.** In parasites lacking TgMIC2, the nuclei do not change shape during 3D motility and the trajectories are less straight than those of wild-type parasites.

the ligand to which TgAMA1 binds. Neither TgAMA1 (*Bargieri et al., 2013*; *Mital et al., 2005*) nor TgRON2 (RVS and GEW, unpublished data) plays any detectable role in parasite motility. Instead, TgMIC2 – another microneme protein secreted onto the parasite surface – likely plays an analogous role during motility to that of TgAMA1 during invasion, with TgMIC2 binding to ligands in the extracellular matrix rather than to TgRON2 in the host cell membrane.

As the parasite moves forward during both invasion and 3D motility, it simultaneously rotates around its long axis (*Leung et al., 2014*; *Håkansson et al., 1999*; *Pavlou et al., 2018*). Our results provide a possible explanation for this rotational motion. In both cases, parasite receptors bind to their cognate ligands in the form of a circular ring of attachment that is fixed in position relative to either the host cell (in the case of invasion) or the extracellular matrix (in the case of 3D motility). It has long been proposed that the TgMyoA motor driving invasion and motility is arranged helically within the parasite, along the spiraling subcortical microtubules (e.g. *Pavlou et al., 2020*). Alternatively, a recent modeling study suggests that the unique geometry of the parasite periphery can generate a self-organized helical pattern of anterior-to-posterior actin flow (*Hueschen et al., 2022*). In either case, if the actin filaments flow helically and are connected to the tails of transmembrane adhesins engaged circumferentially with the host cell surface or extracellular matrix, the forward motion of the parasite either into the cell or through the matrix will by definition be accompanied by rotational twist. Some portion of the perpendicular force that we observed directed towards the annular constriction in our traction force assays may in fact reflect torque on the matrix produced by the parasite as it twists through this circular zone of attachment.

As a parasite invades a host cell, antibodies are 'shaved' off the parasite surface at the moving junction by an as yet unknown mechanism (*Dubremetz et al., 1985*). Remarkably, a similar phenomenon is observed during 3D motility: fluorescent antibody against the surface protein TgSAG1 is lost from the parasite surface anterior to the constriction and shed as a helical trail of fluorescence behind the moving parasite (*Figure 5—figure supplement 2* and *Video 10*).

The numerous parallels between invasion and 3D motility noted here suggest that the two processes may be more mechanistically similar than previously recognized, with implications for our understanding of each. For example, our data suggest that 3D motility can be best described as the parasite undergoing sequential invasion-like events through circular zones of attachment to the

matrix; during invasion the parasite only needs to move a single body length to penetrate into the host cell, but during motility the parasite can string multiple constriction events together to move multiple body lengths (*Figure 5C*). The data also raise the possibility that the characteristic constriction of the parasite seen at the moving junction during invasion does not reflect the parasite being squeezed as it *pushes* itself through the small diameter opening into the host cell, as is commonly thought (e.g. *Herm-Götz et al., 2002*; *Mordue et al., 1999*), but rather the parasite attaching to and *pulling* on the host cell plasma membrane as it penetrates.

## The linear motor model

The data presented here with wild-type parasites are at least partially consistent with the linear motor model of motility, in which: (a) actin polymerization is nucleated at the apical tip of the parasite by formin1 (*Tosetti et al., 2019*); (b) TgMyoA translocates the actin filaments in an anterior-to-posterior direction along the length of the parasite; and (c) binding of the rearward flowing actin filaments to transmembrane adhesins (*Jacot et al., 2016*), whose extracellular domains are attached to the substrate, drives the forward motion of the parasite (*Figure 1A*). While anterior-to-posterior actin flux may occur along the entire length of the parasite periphery (*Hueschen et al., 2022*; *Quadt et al., 2016*; *Stadler et al., 2017*), we propose a modification to the linear motor model in which force is generated, in whole or in part, by the rearward translocation of the subset of actin filaments that are coupled to adhesins at the circular ring of attachment. We were unable to detect any anterior-to-posterior bias in the force vectors at the constriction, which would be expected if the parasite pulls against this zone of attachment in order to move forward. These forces may be below our limit of detection (~0.72 pN). Alternatively, compression of the fibrin immediately adjacent to the constriction might increase the local elastic modulus of the matrix, making it more difficult to see proximal displacements because of a local increase in the displacement detection threshold. Because the fibrin gel behaves primarily as an elastic matrix (*Figure 3—figure supplement 1*), the relationship between acceleration and force is complex and we cannot calculate with any confidence how much total matrix displacement / force would be required to achieve a given parasite velocity and relate this to the displacement detection threshold of the assay.

The regularity with which the parasite generates the circular zones of attachment on its surface, yielding on average one constriction per body length of motion (*Figure 5C*), could be due to periodic bursts of formin1-mediated actin polymerization at the parasite's apical tip (*Tosetti et al., 2019*) or to periodic release of adhesins such as TgMIC2 from the micronemes. Alternatively, the observation that constrictions occur back-to-back almost 50% of the time (*Figure 5A*), may suggest that a new constriction forms at the apical tip before the prior constriction has ended, but this new constriction stalls as the TgMyoA (or possibly TgMyoH *Tosetti et al., 2019*) to which it is connected pulls against the adhesive forces of the more posterior previous constriction. Future real-time analysis of actin polymerization and microneme secretion in motile parasites, together with higher sensitivity force mapping, will be necessary to further interrogate the periodic nature of ring assembly.

The lack of a motility-associated constriction in TgMyoA-deficient parasites could reflect an inability either to assemble ring of adhesins at the apical tip of these parasites or, more likely, to move through the incipient ring once it has formed. Alternatively, it could be that the ring-shaped structures that form during motility are contractile, and contractility requires TgMyoA. The inward deformation of the fibrin matrix towards the constriction reported here is consistent with a contractile ring, as is the physical constriction of the parasite plasma membrane. Intriguingly, the speed of the parasite's forward motion increases once the constriction passes the halfway point on the parasite's longitudinal axis (*Figure 5—figure supplement 1*). This observation suggests that physical squeezing of the tapering posterior end of the parasite by the narrow (and potentially contractile) constriction may contribute to the parasite's forward motion.

If the coupling of actin flow to the circular zone of attachment is important for forward motion, how do the parasites lacking TgMIC2 that move do so at near normal speeds (*Gras et al., 2017*) in the absence of constrictions? First, it is possible that both the wild-type and mutant parasites generate small forces distributed over their entire surface (in addition to larger forces that may be generated at the constriction in wildtype), which are collectively sufficient to drive motility but individually below our level of detection. Alternatively, the loss of TgMyoA or TgMIC2 might lead to compensatory changes in the expression of other genes with overlapping or redundant function (*Meissner et al., 2013*;

*Frénal and Soldati-Favre, 2015*; *Lamarque et al., 2014*). The TgMIC2 KO parasites do in fact show changes in the expression of several other microneme and motility-associated proteins (*Gras et al., 2017*). Finally, it is possible that parasites lacking TgMyoA and/or TgMIC2 use a motility mechanism that is entirely different from that of wild-type parasites (*e.g.*, *Egarter et al., 2014*; *Gras et al., 2019*) and does not involve a constriction but is nevertheless capable of supporting motility and sustaining the parasite's lytic cycle. By analogy with observations from animal cells (*Lämmermann et al., 2008*), parasites might use different mechanisms for motility in different situations such as squeezing through tight junctions (*Barragan et al., 2005*) *vs.* migrating through loose interstitial tissues. If the motility of the mutants is driven by a different mechanism than that of wild-type parasites, higher sensitivity 3D traction force mapping may reveal informative differences in their pattern of force generation.

### A guidance system for motility?

The forward motion of the TgMIC2 knockout parasites, which lack the circular attachment zone, is significantly more disorganized than the motility of wild-type parasites. The mutant parasites can still generate relatively long trajectories, but the trajectories take them less far from their starting point than wild-type parasites. As discussed above, attachment to the matrix via a circular ring of adhesion likely contributes to the rotational twisting of the parasite as it moves forward and thereby to the helicity of the parasite's trajectory. Extensive studies of the helical swimming behavior of bacteria have shown that a curved cell shape and helical trajectory are a particularly efficient way for small cells to move through viscous media (*Berg and Turner, 1979*; *Ferrero and Lee, 1988*; *Hazell et al., 1986*; *Kaiser and Doetsch, 1975*; *Kimsey and Spielman, 1990*). Engagement of *T. gondii* with the extracellular matrix through a circular band of adhesion may therefore not only create a fixed platform for the parasite to pull against and move forward; it may also function as part of a guidance system to help the parasite move efficiently through the various environments it encounters as it disseminates through the tissues of the unfortunate organisms it infects.

## Methods

### Parasite and cell culture

Parasites (RH strain, unless otherwise noted) were propagated by serial passage in human foreskin fibroblasts (HFFs; American Type Culture Collection #CCD-1112sk). HFFs were grown to confluence in Dulbecco's Modified Eagle's Medium (DMEM) (Life Technologies, Carlsbad, CA) containing 10% (vol/vol) heat-inactivated fetal bovine serum (FBS) (Life Technologies, Carlsbad, CA), 10 mM HEPES pH 7, and 100 units/ml penicillin and 100 µg/ml streptomycin, as previously described (*Roos et al., 1994*). Prior to infection with *T. gondii*, the medium was changed to DMEM supplemented with 10 mM HEPES pH 7, 100 units/ml penicillin and 100 µg/ml streptomycin, and 1% (vol/vol) FBS. TgMIC2 knockout parasites (*Gras et al., 2017*) and parasites conditionally depleted of TgMyoA (*Egarter et al., 2014*) were generously provided by Dr. Markus Meissner; the identity of these lines was authenticated by western blot to confirm the absence of the relevant proteins. HFFs and all parasite lines were tested periodically for *Myocoplasma* contamination.

### Pitta chamber assembly

22×22 coverglasses were washed with Alconox detergent, rinsed with tap water, deionized water, and ethanol (100%), and air dried. Two strips of double-sided tape (Scotch 3 M, St. Paul, MN) were placed 3 mm apart on a glass slide, and the coverglass was placed on top of the tape and pressed firmly to ensure a complete seal. The flow cell volume was approximately 10 µl.

### Matrigel matrix

Pitta chambers containing parasites embedded in polymerized Matrigel were prepared as previously described (*Leung et al., 2014*). Briefly, parasites were harvested from infected HFFs via syringe release and passed through a 3 µm Nucleopore filter (Whatman, Piscataway NJ). Parasites were pelleted and resuspended at in Live Cell Imaging Solution (LCIS) buffer. For fluorescence imaging, the LCIS contained 0.5 mg/ml Hoechst 33342 (Thermo Scientific, Waltham, MA) and/or a 1:20 dilution of Alexa546-conjugated anti-TgSAG1 (see *Table 1*). Fluorescent anti-TgSAG1 (100 µg) was prepared using the Alexa Fluor 546 Antibody Labeling Kit (Thermo Scientific, Waltham, MA) as per

**Table 1.** Imaging parameters for the different experiments described.

| Experiment | Objective | Fluorochrome (Excitation/emission wavelengths) | Image spacing in z | Exposure time per image | Number of Image stacks | Total time | Volume (x, y, z) |
|---|---|---|---|---|---|---|---|
| Microspheres (Matrigel) | 60× | DragonGreen (490/507–530 nm) | 41 slices, 1 µm apart | 16ms | 60 | 64 s | 225.3 µm × 84.5 µm×40 µm |
| | | tdTomato parasites (550/579–608 nm) | | | | | |
| Fibrin vs Matrigel and TgMIC2 KO directionality | 20× | Hoechst 33342 (385/420–449 nm) | 41 x 1 µm | 16ms | 120 | 80 s | 665.6 µm × 249.6 µm×40 µm |
| Force Mapping, WT (Fibrin) | 60× | tdTomato parasites (550/579–608 nm) | 50x0.5 µm | 16ms | 60 | 96 s | 225.3 µm × 84.5 µm×24.5 µm |
| | | Fibrin (635/666–723 nm) | | | | | |
| WT, TgMyoA KO, TgMIC2 KO; Brightfield (Matrigel, fibrin) | 60× | N/A: Brightfield | 21 x 1 µm | 40ms | 360 | 302 s | 225.3 µm × 225.3 µm×20 µm |
| Nuclear size vs constriction (Matrigel) | 20× | Hoechst 33342 (385/420–449 nm) | 41 x 1 µm | 16ms | 60 | 80 s | 665.6 µm × 249.6 µm×40 µm |
| | | Anti-SAG1 Alexa 548 (550/579–608 nm) | | | | | |
| TgMyoA KO Force Map (Fibrin) | 60× | Hoechst 33342 (385/420–449 nm) | 50x0.5 µm | 16ms | 60 | 96 s | 225.3 µm × 84.5 µm×24.5 µm |
| | | Fibrin (635/666–723 nm) | | | | | |
| TgMIC2 KO Force Map (Fibrin) | 60× | YFP cytosol (490/507–530 nm) | 50x0.5 µm | 16ms | 60 | 96 s | 225.3 µm × 84.5 µm×24.5 µm |
| | | Fibrin (635/666–723 nm) | | | | | |

the manufacturer's instructions. Parasites were incubated with the Hoechst 33342 (10 min) and/or anti-TgSAG1(15 min) at room temperature, then mixed with Matrigel and LCIS on ice in a 1:3:3 (vol/vol/vol) ratio and immediately added to a Pitta chamber. The Pitta chamber was incubated for three minutes at 35 °C in a custom-build heated microscope enclosure (UVM Instrumentation and Model Facility, Burlington, VT) before imaging.

## Fibrin matrix

Unlabeled and Alexa-Fluor 647-labeled fibrinogen (both from Thermo Scientific, Waltham, MA) were each dissolved in 0.1 M sodium bicarbonate (pH 8.3)–15 mg/ml. Aliquots of 50 µl were flash frozen and stored at –80 C until use. Fibrin gels (*Owen et al., 2017*) containing parasites were prepared in Pitta chambers as follows. The parasite culture medium was replaced with LCIS buffer before harvesting via syringe release and filtering. A total of 500 µl of the parasite suspension were pelleted and resuspended in LCIS buffer to achieve a higher parasite concentration. Parasites were mixed with fibrinogen (final fibrinogen concentration of 2.25, 4.5, or 9 mg/ml). LCIS buffer containing thrombin (Sigma, Burlington, MA) and FBS was added to the parasite-fibrinogen suspension for final concentrations of $1.5 \times 10^8$ parasites per ml, 1 unit/ml of thrombin, and 10% FBS. The mixture was immediately pipetted into a Pitta chamber and allowed to polymerize at room temperature for three minutes before mounting on the microscope and imaging.

A Nikon A1R-ER point-scanning confocal microscope was used to visualize the porosity of the fluorescent fibrin matrix (*Figure 2A* and *Video 4*), Using Galvano scanning, image (1024×1,024 pixel) stacks were captured with 60×Apo (0.1 µm/pixel, NA 1.49 l) objective with 0.25 µm spacing over 25 µm. The LUNV laser was used at wavelength 633 nm and the pinhole size was 28.10 µm.

## Image acquisition

All other imaging was done on a Nikon Eclipse TE300 widefield epifluorescence microscope (Nikon Instruments, Melville, NY) equipped with a NanoScanZ piezo Z stage insert (Prior Scientific, Rockland, MA). See *Table 1* for details of the imaging setup. Time-lapse video stacks were collected with an iXON Life 888 EMCCD camera (Andor Technology, Belfast, Ireland) using NIS Elements software v.5.11 (Nikon Instruments, Melville, NY). Fluorescently labeled parasites were imaged using a pE-4000 LED illuminator (CoolLED, Andover England) and a 89402 Quad filter (Chroma, Bellows Falls, VT). Stacks consisting of individual images (1024 pixel ×384 pixel) captured 0.5–1 μm apart in *z*, covering a total of 10–40 μm, were collected using either a 20×PlanApo $\lambda$ (0.65 pixel/μm, NA 0.75) or 60×PlanApo $\lambda$ (0.22 pixel/μm, NA 1.4) objective as described in *Table 1*. The same volume was successively imaged 60–360 times over the course of 64–302 seconds. The camera was set to trigger mode, no binning, readout speed of 35 MHz, conversion gain of 3.8 x, and EM gain setting of 300.

## Tracking parasite motility

Datasets were analyzed in Imaris ×64 v. 9.2.0 (Bitplane AG, Zurich, Switzerland). Fluorescently labeled parasite nuclei were tracked using the ImarisTrack module within a 1018 pixel ×380 pixel region of interest to prevent artifacts from tracking objects near the border. Spot detection used an estimated spot diameter of 3.0×3.0 × 6.0 μm (*x, y, z*). A maximum distance of 6.0 μm and a maximum gap size of 2 frames were applied to the tracking algorithm. Tracks with durations under 10 seconds or displacements of less than 2 μm were discarded to avoid tracking artifacts and parasites moving by Brownian motion, respectively (*Leung et al., 2014*). Accurate tracking was confirmed by visual inspection of parasite movements superimposed on their calculated trajectories. All 3D trajectory analysis was done using data from three biological replicates, each consisting of three technical replicates. Student's t-tests were used to determine statistical significance between samples.

## Fibrin deformation with FIDVC

The Fast Iterative Digital Volume Correlation (FIDVC) algorithm was used to calculate 3D fibrin displacements (*BarKochba et al., 2015*) by comparing two consecutive image volumes. The image volumes were cropped to 384×384 pixels (*x,y*)×48 slices (*z*) before running FIDVC. The initial interrogation window size was set to 32×32 × 32 and the program was run incrementally. The displacements arrows were plotted using a combination of previously described MATLAB code for 2D quiver color-coding for directionality (*Owen et al., 2017*) and custom MATLAB code (see accompanying source code file). To determine the displacement detection threshold, we calculated the mean magnitude of the 16,807 displacement vectors for each time point comparison in each dataset. The displacement detection threshold for that dataset was set as three standard deviations above the lowest of these mean values.

## Rheology

A Lumicks C-Trap laser trap was used to determine the viscoelasticity of the fibrin. First, the power density spectrum (PDS) was calculated with 0.91 μm styrene beads in a flow cell. The beads were embedded in a fibrin gel, and a bead was captured in the laser trap. The flow cell was oscillated on the stage in the *y* dimension at different frequencies (1, 5, 10, 20, 50, 100 Hz), each for 5 s, at a fixed amplitude of 100 nm. The trap position and force in the *y* dimension were captured at 78 kHz. The elastic (in-phase) and viscous (out-of-phase) moduli were calculated using the bead's force trace. The data were analyzed in R studio. A sliding two-sided window filter, size 78, was applied to both traces. The position signal was fitted with the function $position = A*\sin(t*f - phase)$ using nonlinear least-squares with fit parameters for amplitude (A), frequency (f), and phase. The force readout was fit with the function $force = B*\sin(t*f - phase) + C*\cos(t*f - phase)$ using nonlinear least-squares with fitting parameters B and C, where B is the in-phase (elastic) component of the force. The C fit parameter divided by the velocity of the trap equals the viscous modulus.

## Acknowledgements

This work was supported by U.S. Public Health Service grants: AI139201 and AI137767 (GEW): GM141743 and S10OD026884 for the Lumicks C-Trap (DMW); and T32AI055402 and F31AI145214

(RVS), as well as American Heart Association grant 19PRE34370071 (RVS) We thank Nicole Bouffard and Dr. Douglas Taatjes of the UVM Larner College of Medicine Microscopy Imaging Center (RRID# SCR_018821) for helpful advice and assistance with the Nikon A1R-ER confocal microscope, which is supported by U.S. Public Health Service grant 1S10OD025030-01 from the National Center for Research Resources. We thank: Dr. Markus Meissner for sharing the TgMyoA and TgMIC2 knockout parasites; Dr. David Sibley for anti-TgSAG1 antibody; Drs. Mark Rould, Alex Dunn, Christina Hueschen, Li-av Segev Zarko, John Boothroyd, Leanna Owen, Markus Meissner, and Ulrich Schwarz for helpful discussions; Anne Snyder, Drs. Robyn Kent, Christina Hueschen, and Chris Huston for helpful feedback on the manuscript. Finally, we thank Dr. Christian Franck for making the FIDVC code used in this work freely available through GitHub.

## Additional information

### Funding

| Funder | Grant reference number | Author |
| --- | --- | --- |
| National Institute of Allergy and Infectious Diseases | AI139201 | Gary E Ward |
| National Institute of Allergy and Infectious Diseases | AI137767 | Gary E Ward |
| National Institute of General Medical Sciences | GM141743 | David M Warshaw |
| National Institute of General Medical Sciences | S10OD026884 | David M Warshaw |
| National Institute of Allergy and Infectious Diseases | T32AI055402 | Rachel V Stadler |
| National Institute of Allergy and Infectious Diseases | F31AI145214 | Rachel V Stadler |
| American Heart Association | 19PRE34370071 | Rachel V Stadler |

The funders had no role in study design, data collection and interpretation, or the decision to submit the work for publication.

### Author contributions

Rachel V Stadler, Conceptualization, Data curation, Software, Formal analysis, Funding acquisition, Validation, Investigation, Visualization, Methodology, Writing – original draft; Shane R Nelson, Software, Formal analysis, Validation, Investigation, Visualization, Methodology, Writing - review and editing; David M Warshaw, Conceptualization, Resources, Formal analysis, Supervision, Funding acquisition, Validation, Visualization, Methodology, Project administration, Writing - review and editing; Gary E Ward, Conceptualization, Resources, Data curation, Formal analysis, Supervision, Funding acquisition, Validation, Visualization, Methodology, Writing – original draft, Project administration

### Author ORCIDs

Rachel V Stadler http://orcid.org/0000-0002-1049-1638
Gary E Ward http://orcid.org/0000-0003-4138-3055

### Decision letter and Author response

Decision letter https://doi.org/10.7554/eLife.85171.sa1
Author response https://doi.org/10.7554/eLife.85171.sa2

## Additional files

### Supplementary files
• MDAR checklist

• Source code 1. The source code generates 2D and 3D quiver plots of the FIDVC data showing either all vectors or only those vectors above a background threshold.

## Data availability
The FIDVC code used for the 3D force mapping is freely available through github (https://github.com/FranckLab/FIDVC, copy archived at swh:1:rev:446a88aaa9e62e8a4744e7165a18ed1bdb0c7c1a).

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
