## [Editor Report]

The authors report the biophysics behind *Toxoplasma gondii* locomotion on 3D matrixes that are fluorescently labelled and hence allow the detection of their displacements and calculation of force vectors. The authors discover that tachyzoites move by a high degree of continuous constrictions reminiscent of those seen during cell invasion. They probe not only wild type parasites but also two key mutants, which reveal a striking absence of the constrictions and changed trajectories. Due to the uniqueness of eukaryotic gliding motility, its high speed and the importance of infection, this manuscript will be of general interest not only to cell biologists studying cell migration in context of infectious diseases but also appealing to biophysicists looking at cellular force generation.

---

## [Decision Letter]

[Editors' note: this paper was reviewed by Review Commons.]

---

## [Author Response]

We thank the reviewers for their thoughtful comments and suggestions. Our point-by-point responses to the reviewer comments and suggestions are summarized below.

Reviewer #21) I could not open any of the videos (while those associated with the BioRXiv preprint were fine). Some of the videos could be combined to minimize download/open clicking sequences.

The videos were uploaded as.avi files, as per Review Commons instructions, and we tested our ability to view them on several computers at our institution before submission. We will ask the Review Commons Managing Editor to make sure there are no problems with the videos uploaded with the revised manuscript.

2) I really dislike reviewing papers without line numbers

Line numbers have been added to the revised version.

3) The manuscript could be made more relevant to malaria researchers by briefly discussing red cell invasion by merozoites (a single constriction and force against the cell cortex), migration of ookinetes (multiple constrictions during mosquito gut penetration) and sporozoites (long distance migration), but this is not a must.

Constrictions during ookinete migration are now mentioned on lines 265-269, and the discussion of the constriction at the moving junction has been broadened to include other apicomplexan parasites lines 270-278.

4) I would limit reporting of numbers to two digits, e.g. instead of 46.3% make it 46%; 2.56 +/- 0.38 to 2,6 +/- 0,4 etc

We have adjusted all numbers in the text and figures to the appropriate number of significant figures based on measurement precision.

5) Millions of deaths, please rewrite, more like around 1 million from malaria and cryptosporidium; use citation (WHO)

Done (line 40)

6) Motility: please don't mention flagella, which are used for swimming, in the same sentence / phrase / logic connection as lamellipodia, which are used for substrate based migration

The sentence has been rewritten to make clear that cilia and flagella are not organelles involved in the substrate-dependent motility of other eukaryotic cells (lines 47-49).

7) In Figure 1B, I can see one microsphere and it's not clear if it moves completely back to the original position. In the video it looks like it goes completely back, maybe exchange the last panel of the figure with a last frame from the video? Or maybe better: replace with frames from video 2, which is more striking and shows many beads being displaced?

As suggested, Figure 1B now shows frames from the other video (former Video 2), where bead movement is more obvious.

8) Please add the entire figure S1 to Figure 1. This is important for readers to understand and 'deserves' full figure status. Same for Figure S2.

We have moved most of former Figure S1 into a new main Figure 2, as suggested. We left the two graphs as Supplemental data (new Figure S1), since these graphs simply show that parasite motility in fibrin is similar to the previously described motility of parasites in Matrigel.

Figure S2 has been moved to the main text, as suggested (in new Figures 3 and 6).

9) I would encourage the authors to elaborate more on the data on Figure S2. It appears that motile parasites did mostly not exert forces above the level for non-motile parasites; for how much motility did they observe forces? The meaning of the x-axis does not become clear. Are those individual parasites per time point or time points of one parasite or of the analyzed matrix volumes over several parasites? How many parasites where observed? This is stated more clearly later but needs to be done already here.

We have moved the data in former Suppl. Figure S2 into the main figures, broken it into two parts (Figures 3 and 6B-E) and included a new 3D volume view and additional explanatory detail in the figure legends and text to clarify these points of confusion (lines 100116, 500-507, 564-570).

10) Please change 0.042 μm into 42 nm etc

Done, lines 113-116.

11) Please move some of the data in Figure S8 to the main figures e.g. Figure 4, where it would make a nice contrast / comparison to the mic2 mutant. Please also put a WT for comparison.

Done; see revised Figure 6.

12) I wonder if the defect in directional migration of the mic2 mutant is also partly due to the parasite not being able to squeeze through narrow matrix pores and hence is deflected more often. While I understand (and agree) with the authors observation (interpretation) of the wt parasites not squeezing but pulling, it's hard to think that such squeezing would not still play a part.

The idea that the parasite needs to squeeze its way through pores in the matrix is intuitively appealing (and, in fact, what we had expected to see) but there is currently no data to support it. If squeezing were occurring, we should see an outward deformation of the matrix as the parasite pushes on the matrix fibers, but this is something we have never observed. We therefore think it is unlikely that the loss of directional migration is due to an inability to squeeze through pores in order to “stay on track”.

13) Hueschen et al. is now on BioRXiv

The BioRXiv citation has been added (lines 293, 320).

14) The shaving off of antibodies could be brought into context to the work on sporozoites by Aliprandini Nat Micro 2018 and on trypanosomes by Enstler Cell 2007 (but not a must)

The two studies mentioned are intriguing and may be related to the well-documented anterior to posterior flux and shedding of GPI-anchored proteins from the surface of gliding Toxoplasma tachyzoites. What we are showing here is slightly different: the fluorescent antibodies on the cell surface seem to be “shaved” backwards at the constriction, much like surface bound antibodies are shaved backwards at the moving junction during invasion (Dubremetz 1985). In other words, there is a discontinuity in the density of surface staining at the constriction/junction. All of these processes may be related, but this is only speculation at this point and since the shaving of antibody at the constriction is a minor point of the paper (meant only to illustrate another similarity between 3D motility and invasion), we would prefer not to try to tie it to these other observations which may or may not be related.

15) Anterior-posterior flux: best experimental evidence for this is Quadt et al. ACS Nano 2016 for Plasmodium and Stadler MBoC 2017 for Toxoplasma. The common observations and differences could be discussed as they pertain to the current study

These two papers are now cited in our discussion of the linear motor model along with our speculation that the constriction reflects the motility-relevant zone of engagement of this rearward flux with ligands in the matrix (lines 319-322).

16) The loss of mic2 could lead to the loss of the capability to form discrete adhesion sites that reveal themselves as the observed rings in 3D. I suggest to be careful to hypothesize that the absence of this and MyoA reveals a completely different motility mechanism. To me it seems more likely that the absence of the proteins means that the existing mechanism doesn't work perfectly any more, ie the highly tuned migration machinery misses a key part and malfunctions.

The paragraph in question offered possible explanations for how parasites lacking the constriction could in fact move at normal speeds, not that motility was negatively affected. We have tried to make this more clear in the revision (lines 352-354), before describing the 3 possible explanations.

17) Maybe reflect on whether 'search strategy' might be a better word than 'guidance system'

We have replaced the term “guidance system” in the title (lines 1-2), abstract (lines 3336) and introduction (line 75) with more conservative references to the ability of the parasite to move directionally. The only place the term “guidance system” remains is in the final paragraph of the discussion, which is more speculative in nature, and where we now suggest it to be “part of” a guidance system.

Reviewer #31) Extracellular matrix choice. The authors track the parasite movement first on Matrigel and next on fibrin. The authors exemplify the fibrin matrix on an image on Suppl. Figure 1 that shows a relatively quite large pore size, similar or greater than parasite size. Was the analysis done on parasites touching the fibers?

Previous Suppl Figure 1A showed a confocal image at only one z-plane which did indeed give the impression that the pores are relatively large. We have changed this image to a more informative maximum intensity projection (New Figure 2A) and included a video showing the entire imaging volume (new Video 4), which makes clear that the matrix contains many small fibers and that the pores are smaller than the previous single z-plane suggested, so the parasite is likely to be near to or in contact with fibers of the matrix at all times. In Suppl Figure 1D we purposely used a less dense matrix in order to make the matrix deformation more obvious to the eye. The density of the matrix in Figure 1D has been added to the legend.

2) Lack of movement of parasites. In many figures of the articles it is revealed that the majority of parasites in fibrin remain immobile (Suppl Figure 1, Figure 2, Video 5, Suppl Figure 2, Suppl Figure 8). The number of immobile parasites in Matrigel seem to be lower than in fibrin (Suppl Figure 1B) although no quantification is shown. How does the movement in fibrin and Matrigel compare? How does this compares with movement in stiff substrates in 2D? Could the lack of movement be caused by the large pore site in fibrin?.

We have added a panel to Suppl. Figure S1 showing that the proportions of parasites moving in fibrin vs Matrigel are not significantly different. In fact, none of our measured motility parameters are different between fibrin and Matrigel. Not all parasites move during the 80s of capture used for these matrix comparisons; some of the parasites are likely dead, but others may have simply not initiated motility during this time window. We typically see between 30-50% movement in 3D motility assays of this duration and similar numbers in 2D trail assays although we have not explored the effect of 2D substrate stiffness.

3) Considering parasite movement: The authors consider that 3SD is a cutoff for considering parasite displacement. However, several timepoints fall behind this cutoff in the control without parasites and the knockouts with restricted movement.

We chose three standard deviations from the mean as our cutoff, in order to eliminate 99.7% of the noise. Since we calculate 16807 vectors per comparison, this leaves us with ~50 vectors above the cutoff even in samples with no moving parasites. Not surprisingly, these vectors are found at random locations in the volume. New Figures 3 and 6B-E and the associated text (lines 100-116, 500-507, 564-570) hopefully clarify this point adequately; it is quite obvious in Figure 3C which vectors correspond to parasite-induced displacements and which correspond to random noise.

4) Imaging: Although the authors show a very detailed an illustrative table of the imaging acquisition conditions in table 1, it is unclear which microscope the authors used, as two microscopes are described in the methods section, a Nikon Eclipse TE300 widefield microscope and a Nikon AIR-ER confocal microscope. Which images were taken in each system? For the location of Table1 in the manuscript it seems that most images were taken with the Nikon Eclipse. Although this microscope has control over z, the images are quite noisy. How does the lack of confocallity might interfere with the analysis?

The high temporal resolution needed for 3D force mapping of cells that move several microns per second meant that all these experiments were done using a widefield microscope equipped with a piezo-driven z-stage. The fastest confocal we tested was not as fast as the widefield. However, spatial resolution suffered as a result of having to use widefield, particularly in z, and this did indeed make our data more noisy as suggested by the reviewer. This may be why we were unable to detect fibrin deformation in the knockout parasites. The only data collected on the confocal microscope were those shown in new Figure 2A; we have clarified this on lines 421-427. Future studies will explore other imaging modalities such as light sheet microscopy in an attempt to achieve better spatial resolution while maintaining the high frame rates required for force mapping.

5) Nuclear constriction. The authors did not show any image or video exemplifying this.

The images in Suppl. Figure 6 have been replaced with data that show the nuclear shape more clearly.

6) Knockouts: The authors did not explain how did they generated the knockouts in the methods or did now show the efficacy of the knockout in any figure. If these knockout strains were a gift (I did not find it on the manuscript), the authors should indicate this more explicitly and reference the manuscript where they were described for the first time.

Both of the stable knockout lines used were generous gifts from Dr. Markus Meissner. We cited the original papers describing these lines in the text and thanked Dr. Meissner for providing them in the Acknowledgements section. We have now included an additional citation at the first mention of each of the knockouts (lines 174, 188) to make it even clearer where they came from.

7) Discussion: Although the experimental methodology is sound the authors seem to make many assumptions and speculations on the discussion as how the appearance of this ring/constriction on the parasite translates into the helical movement of the parasite or the coupling of the ring with the cytoskeleton. Live imaging of actin dynamics or mathematical modelling could be used to support their claims.

We imaged parasites expressing the actin chromobody but were unable to visualize a ring of actin at the constriction. However, due to the speed of the parasites and the need for a fast frame rate (~15 ms per image) to reconstruct the 3D image volumes, the actin chromobody signal could be under our threshold of detection. We need to develop new, more sensitive ways to visualize proteins at the constriction, and this will be a major focus of our work going forward. > We fully concur that mathematical modeling such as the work recently done by Hueschen et al. on actin flow during motility and by Pavlou et al. on the role of parasite twist during invasion has much to offer our understanding of these processes. Similar approaches may provide support to the speculations (not claims!) we offer in the discussion and, although beyond the scope of the current study, are a direction we intend to take this work in the future – particularly if we are able to improve the signal-to-noise in our force mapping.

8) Quantification of experiments missing: Overall, the main figures lack quantification that sometimes can be found in the supplemental information and sometimes is missing. I would suggest including quantifications next to the events described in the main figures). Likewise, some of the supplemental figures lack quantification (Suppl Figure 7, how many parasites showed this protein trail?)… Overall, the authors should indicate how many parasites were quantified in each figure. As they usually refer to number of constrictions. This is overall a problem in main figures 3 and 5. Or for example in Suppl Figure 5: How many parasites were quantified in this figure? The authors only show number of constrictions, and as the authors described, a parasite might have more than one constriction.

We have added further detail on the number of events/parasites quantified to both the figure legends and text throughout the manuscript, including the specific examples noted by the reviewer.

9) Videos: The videos lack scale of time. Although this that can be found in main figures, it would be helpful to have the annotation in the videos. Likewise, some references for positions in videos, such as the cross found on Figure 1 would be helpful for parasites that present little movement.

Time stamps have been added to all videos as suggested, and crosshairs have been applied to new Figure 1B and Suppl. Figures 7 and 8 to make the movement of the parasites more obvious.

Reviewer #41) I am not sure about the premise that the "linear model" of gliding motility predicts uniformly forward direction. Previous videos of 2D gliding show sporadic motility, changes in direction, or even reversal of direction are not infrequent. However, the current model could explain these behaviors if one or more of the following conditions occur: 1) myosin motors might be coordinating activated to initiate motility, followed by relaxation, 2) actin fibers might be transiently arrayed in clusters that change density and polarity over time, or 3) adhesins, necessary to generate traction, might vary in density and spatial orientation across the surface of the parasite. Changes in these properties would be expected result in zones that promote or disfavor local forces needed for motility – and reversal of direction could occur when forward forces relax and external elastic forces predominate.

The potential explanations offered by the reviewer for the frequent changes in direction of zoite motility are intriguing and worth exploring experimentally. The ability of actin fibers to periodically reverse polarity, or the presence of counteracting elastic forces are not components of the “standard” linear motor model of motility but, if they occur, could explain the patch gliding phenomenon and help refine our understanding of motility. Since the data in this manuscript do not in the end either strongly support or disprove the linear motor model – this may ultimately require higher resolution force mapping methods that can detect the forces responsible for forward motion – we have de-emphasized potential problems with the model in the introduction and deleted specific discussion of patch gliding as one of these problems (lines 61-64).

2) The model favored here: "we propose that force is generated, at least in part, by the rearward translocation of the subset of actin filaments that are coupled to adhesins at the circular ring of attachment" does not seem fundamentally different from the current model – other than it focuses the forces at a critical junction that the parasite migrates through. It seems to me that this is a refinement of the current model and not a replacement. As such, the authors might focus on how their data improve the model rather than pointing out prior deficiencies (although I get that editors like this style).

We agree with the reviewer and have modified the text to be more circumspect on this issue (lines 319-331).

3) The finding that the absence of MIC2 affects the constriction formed by inward pull on the matrix is quite convincing and interesting. However, mutants that cannot form the constriction, still move at similar speeds. This suggest that the inward force is different from the motor itself and affects its ability to impart direction, rather than the ability to move per see. The interpretation of the MyoA defect is complicated since motility is certain to be disrupted, the potential role of an independent inward force may no longer be detectable.

We agree with the reviewer on this point as well: the forces we have observed to date cannot explain forward motion. We stated this previously and have now emphasized the point further (lines 322-324, 352-357). Because the parasite is moving forward, the forces responsible must be there but are likely below our threshold of detection. In order to visualize these forces, we are going to need new imaging modalities that can achieve better signal-tonoise than our current setup at the high frame rates required for force mapping. That said, we new data we have added to the manuscript are at least consistent with the narrow diameter ring of the constriction making a contribution to the parasite’s forward motion (new Suppl. Figure 10 and lines 347-351)

4) Although I agree with the authors that there are striking parallels between motility in 3D and cell invasion, I am not certain about their conclusion that the construction seen during cell entry is due to the parasite pulling inwardly. When entering the host cell, the parasite must also navigate the dense subcortical actin network, which likely also aids in forming the constriction that is observed. It would be interesting to record this pattern under conditions where host cell actin is destabilized while parasite motility is intact- for example using cytochalasin D to treat wild type host cells during invasion by resistant parasites.

We do not conclude that the constriction during invasion is due to the parasites pulling inwardly, but we do propose that this possibility needs to be considered based on the noted similarities between invasion and motility and our clear (and somewhat surprising) demonstration that the moving parasite pulls on the matrix at the constriction during motility. During invasion, the parasite may indeed have to squeeze through the dense subcortical network – or it may use secreted proteins to loosen up the network so that no squeezing is required. We just don’t know, and our purpose here was simply to put this alternative possibility on the table because we believe it is a viable possibility that follows from the data presented.

We thank the reviewer for the suggestion of testing what happens when cytoD resistant parasites invade in the presence of cytoD; this is a clever idea that we will likely pursue in future work.

5) Not all of the color patterns shown in Figure 1A are consistent with the model. For example, GAP40 (yellow) does not appear in the model, there are two MLC boxes, but they are different shades, and ELC1/2 does not appear in the model.

We thank the reviewer for catching this error; it has now been fixed.